# *Shigella* type-III secretion system effectors counteract the induction of host inflammation and cell death

Hiroshi Ashida [ID] [1,2 ✉], Tokuju Okano [ID] [1], Tamako Iida[1], Poramed Onsoi [ID] [1], Chihiro Sasakawa [ID] [2] &
Toshihiko Suzuki [ID] [1 ✉]

## Abstract

**Many enteric bacterial pathogens deliver virulence effectors to counteract host innate immune responses, such as inflammation and cell death, and colonize the intestinal epithelium. However, host cells recognize the disruption of their innate immune signaling by bacterial effectors and induce alternative immune responses, collectively termed "effector-triggered immunity", to clear bacterial pathogens. Here, we describe a mechanism of cell death induction via effector-triggered immunity and the bacterial countermeasures of the pathogen *Shigella flexneri*. *Shigella* delivers the OspI effector to inhibit NF-κB activation, which results in caspase-8 activation in return. Deamidation and inactivation of the E2 ubiquitin-conjugating enzyme Ubc13 by OspI results in the inactivation of cIAPs, which serves as a cue to trigger apoptosis and necroptosis. To prevent caspase-8-mediated apoptosis, *Shigella* delivers OspC1 and inhibits caspase-8 activation via its ADP-riboxanation activity, which however triggers necroptosis. Necroptosis induced as a secondary effector-triggered immunity response by OspC1 is eventually prevented by another *Shigella* effector, OspD3. The findings of this study reveal a complex multi-tiered bacterial strategy for circumventing host cell death induction via effector-triggered immunity.**

**Keywords** *Shigella*; Effector; Caspase-8; Apoptosis; Necroptosis
**Subject Categories** Autophagy & Cell Death; Immunology; Microbiology; Virology & Host Pathogen Interaction

## Introduction

Host cells sense bacterial infection and activate innate immune responses, including inflammation and cell death, to clear bacterial pathogens. For example, host pattern recognition receptors (PRRs) detect pathogen-associated molecular patterns (PAMPs), such as lipopolysaccharide (LPS) and peptidoglycan (PGN), or damage- or danger-associated molecular patterns (DAMPs), such as membrane remnants and ATP. Upon detection of PAMPs or DAMPs, PRRs activate inflammatory signaling pathways, including nuclear factor κB (NF-κB), type I interferon (IFN), and the inflammasome. Activation of these pathways induces the expression of inflammatory cytokines, chemokines, and antimicrobial peptide genes that are crucial for bacterial elimination (Vance et al, 2009; Takeuchi and Akira, 2010). Detection of PAMPs or DAMPs also induces several types of cell death, including apoptosis, pyroptosis, and necroptosis (Jorgensen et al, 2017; Broz et al, 2020). Apoptosis is a caspase-dependent, non-inflammatory, non-lytic form of cell death (Kerr et al, 1972). By contrast, pyroptosis is a caspase-1/-4–gasdermin D (GSDMD)-dependent, inflammatory, lytic form, whereas necroptosis is a caspase-independent, but RIPK3-mixed lineage kinase domain-like (MLKL)-dependent, inflammatory, lytic form of cell death (Holler et al, 2000; Degterev et al, 2005; Broz and Dixit, 2016). The elimination of infected cells plays important roles in the clearance of damaged cells, clearance of bacterial pathogens, and presentation of bacteria-derived antigens to the adaptive immune system (Rudel et al, 2010; Yuan et al, 2016; Sundaram et al, 2024). To counteract the induction of innate immune systems by PAMPs or DAMPs, many enteric bacterial pathogens, such as *Shigella*, *Salmonella*, and enteropathogenic *Escherichia coli* (EPEC), deliver virulence proteins called effectors via the type III secretion system (T3SS). Effectors have specific enzymatic activity, and mimic and usurp host cell structure, organelles, or cell signaling to manipulate host innate immune systems and create a replicative niche (Stewart and Cookson, 2016; Scott and Hartland; 2017; Pinaud et al, 2018).

However, PAMPs and DAMPs are not the only mechanisms of pathogen recognition for activation of the innate immune system. In addition to PAMPs and DAMPs, host cells detect the interference with innate immune signaling pathways by bacterial effectors and induce an alternative immune response as a backup host defense mechanism. These anti-pathogen immune responses in which pathogens are identified on the basis of bacterial effector activity are known as "effector-triggered immunity" (Fischer et al, 2020; Remick et al, 2023). For example, the *Yersinia* effectors YopE and YopT inhibit Rho GTPase to manipulate the actin cytoskeleton and block phagocytosis. Host cells specifically recognize the

[1]Department of Bacterial Infection and Host Response, Graduate School of Medical and Dental Sciences, Institute of SCIENCE TOKYO, 1-5-45 Yushima, Bunkyo-ku, Tokyo 113-8510, Japan. [2]Medical Mycology Research Center, Chiba University, 1-8-1 Inohana, Chuo-ku, Chiba 260-8673, Japan. ✉E-mail: ashi.bact@tmd.ac.jp; suzuki.bact@tmd.ac.jp

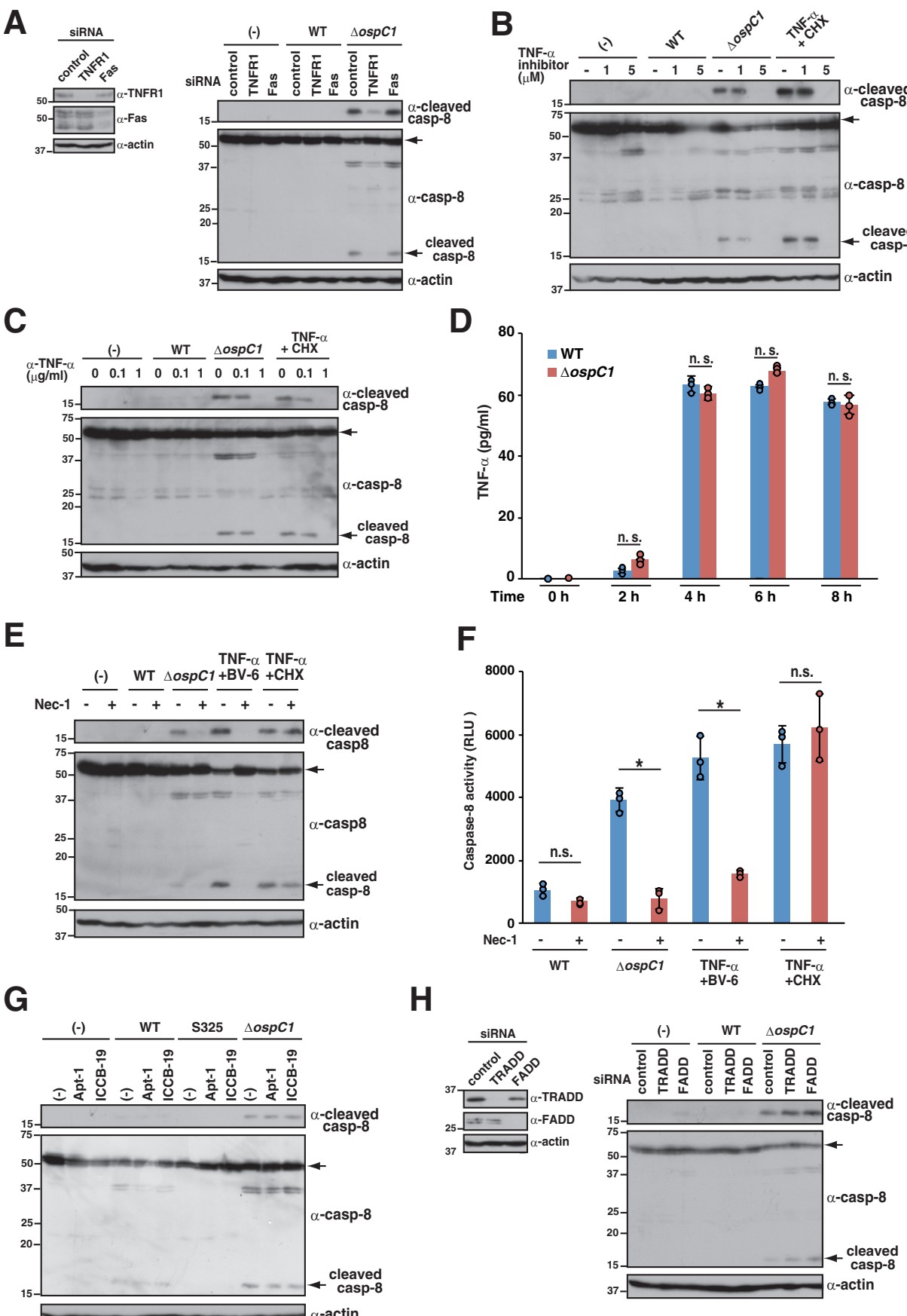

◄ **Figure 1. TNFR signaling is required for caspase-8 activation during *Shigella* infection.**

(A) HT-29 cells treated with control, TNFR1, or Fas siRNAs were infected with the indicated *Shigella* strains and incubated for 8 h. Cell lysates were subjected to immunoblotting. The knockdown efficiency of the indicated siRNAs was assessed by immunoblotting (inset). (B) HT-29 cells were infected with the indicated *Shigella* strains or treated with TNF-α (25 ng/mL) + cycloheximide (CHX; 25 μg/mL) in the presence or absence of a TNF-α inhibitor (1 or 5 μM) and incubated for 8 h. Cell lysates were subjected to immunoblotting. (C) HT-29 cells treated with or without an anti-TNF-α antibody were infected with the indicated *Shigella* strains or treated with TNF-α (25 ng/mL) + CHX (25 μg/mL) and incubated for 8 h. Cell lysates were subjected to immunoblotting. (D) HT-29 cells were infected with the indicated *Shigella* strains. Culture supernatants obtained at the indicated time points were subjected to enzyme-linked immunosorbent assay (ELISA). Data are expressed as the mean ± SD from triplicate and representative of three independent experiments ($P = 0.1183$ for 2 h, $P = 0.3627$ for 4 h, $P = 0.402$ for 6 h, and $P = 0.9959$ for 8 h; two-way ANOVA). (E, F) HT-29 cells infected with the indicated *Shigella* strains or treated with TNF-α (25 ng/mL) + BV-6 (1 μM), or TNF-α (25 ng/mL) + CHX (25 μg/mL) in the presence or absence of Nec-1 (50 μM) were incubated for 6 h. Cell lysates were subjected to immunoblotting (E) or measurement of caspase-8 activity (F). Caspase-8 activity is reported as relative light units (RLU) of infected samples normalized to the value in uninfected samples. Data are expressed as the mean ± SD from triplicate and representative of three independent experiments ($P = 0.9892$ for WT, $P < 0.0001$ for Δ*ospC1*, $P < 0.0001$ for TNF-α + BV-6, and $P = 0.9180$ for TNF-α + CHX; two-way ANOVA). (G) HT-29 cells were infected with *Shigella* WT, S325 (T3SS-deficient mutant), Δ*ospC1* strains in the presence or absence of the TRADD inhibitor Apt-1 (10 μM) or ICCB-19 (10 μM), and incubated for 8 h. Cell lysates were subjected to immunoblotting. (H) HT-29 cells treated with control, TRADD, or FADD siRNAs were infected with the indicated *Shigella* strains and incubated for 8 h. Cell lysates were subjected to immunoblotting. The knockdown efficiency of the indicated siRNAs was assessed by immunoblotting (inset). Data are representative of three independent experiments (A–C, E, G, H). Molecular weights in immunoblots are in kDa. Data are considered significant when $P < 0.05$, with *$P < 0.05$ or n.s., not significant (D, F). Source data are available online for this figure.

inactivation of RhoA by *Yersinia* effectors and activate the pyrin inflammasome and pyroptosis to restrict bacterial infection (Chung et al, 2016; Ratner et al, 2016). In addition, *Yersinia* delivers YopJ, which has acetyltransferase activity and inactivates TAK1, IKKβ, and MKK to reduce inflammation (Mittal et al, 2006; Mukherjee et al, 2006). Host cells sense the activity of YopJ and induce caspase-8–mediated apoptosis or pyroptosis, which suppresses *Yersinia* infection (Peterson et al, 2017; Orning et al, 2018; Sarhan et al, 2018). Therefore, bacterial effectors act as a double-edged sword, providing a niche for bacterial replication by disrupting host immune signaling, and also inducing other host defense systems. Effector-triggered immunity was first characterized in plants as a defense mechanism against pathogenic microbes (Jones and Dangl, 2006; Jones et al, 2016). However, the concept of effector-triggered immunity in eukaryotes was only recently proposed, and the mechanism by which human bacterial effectors trigger such responses and the outcome of effector-triggered immunity remain largely unknown.

*Shigella flexneri* is a highly adapted human pathogen that is capable of invading and colonizing the intestinal epithelium, resulting in severe inflammatory colitis (shigellosis) (Baker and The, 2018; Khalil et al, 2018). *Shigella* infection is sensed by epithelial cells, which produce an inflammatory response and induce cell death as an acute immune defense response to eliminate the bacterial insult. To counteract this, *Shigella* delivers a subset of T3SS effectors to prevent host inflammatory responses and cell death with the aim of maintaining its replicative niche (Ashida et al, 2015; Ashida et al, 2021). For instance, *Shigella* delivers OspC3 and IpaH7.8 effectors to prevent pyroptosis. OspC3 modifies caspase-4 via its ADP-riboxanase activity, resulting in the inhibition of noncanonical inflammasome activation and pyroptosis (Kobayashi et al, 2013; Li et al, 2021). In addition, IpaH7.8 ubiquitinates GSDMD and targets it for proteasomal degradation, thereby inhibiting pyroptosis (Luchetti et al, 2021).

We previously demonstrated that *Shigella* delivers the OspC1 effector to prevent caspase-8-dependent apoptosis (Ashida et al, 2020). Inhibition of caspase-8 activity by OspC1 triggers another form of cell death, necroptosis, as a backup host defense system. In other words, host cells sense the disturbance of caspase-8-apoptosis signaling by OspC1 and induce necroptosis via effector-triggered immunity. *Shigella* responds by delivering another effector, OspD3,

the protease homolog of EPEC EspL, which prevents necroptosis by targeting receptor-interacting serine/threonine-protein kinase 1 (RIPK1) and RIPK3 for degradation (Pearson et al, 2017; Ashida et al, 2020). In this manner, *Shigella* prevents several types of host cell death to promote colonization. However, the stimuli that drive caspase-8 activation and the molecular mechanism by which the *Shigella* OspC1 effector blocks caspase-8 activation to prevent apoptosis remain unclear.

In this study, we provide the first evidence that caspase-8-apoptosis and necroptosis during *Shigella* infection are driven by bacterial T3SS effector activity rather than by PAMPs or DAMPs. The *Shigella* OspI effector inactivates Ubc13 in the NF-κB signaling pathway to modulate host inflammatory responses at the expense of triggering caspase-8-dependent apoptosis. We also demonstrated multilayered *Shigella* strategies that counteracted host cell-death induction as part of effector-triggered immunity. These strategies allow *Shigella* to successfully prevent inflammation, apoptosis, and necroptosis, thereby maintaining its replicative niche.

## Results

### *Shigella ΔospC1* induces caspase-8 activation through TNFR signaling

Caspase-8-dependent cell death is critical for host defense, facilitating clearance of bacterial infections (Orning et al, 2018; Sarhan et al, 2018; Demarco et al, 2020; Roncaioli et al, 2023). The *Shigella* OspC1 effector blocks caspase-8 activation to prevent apoptosis; however, the stimuli that drive caspase-8 activation remain unclear. Hence, we elucidated the molecular mechanism underlying caspase-8 activation during *Shigella* infection. Caspase-8 can be activated by stimulation of cytoplasmic membrane-localized death receptor family proteins, such as TNF receptor 1 (TNFR1) and Fas (Yuan and Ofengeim, 2024; Ai et al, 2024). We therefore treated cells with siRNA targeting TNFR1 or Fas and found that knockdown of TNFR1, but not Fas, in cells infected with Δ*ospC1* decreased the levels of cleaved caspase-8 to the levels observed in cells infected with WT *Shigella* (Fig. 1A). Similarly, treatment with a TNF-α inhibitor or TNF-α-neutralizing antibody prevented caspase-8 cleavage in Δ*ospC1*-infected cells,

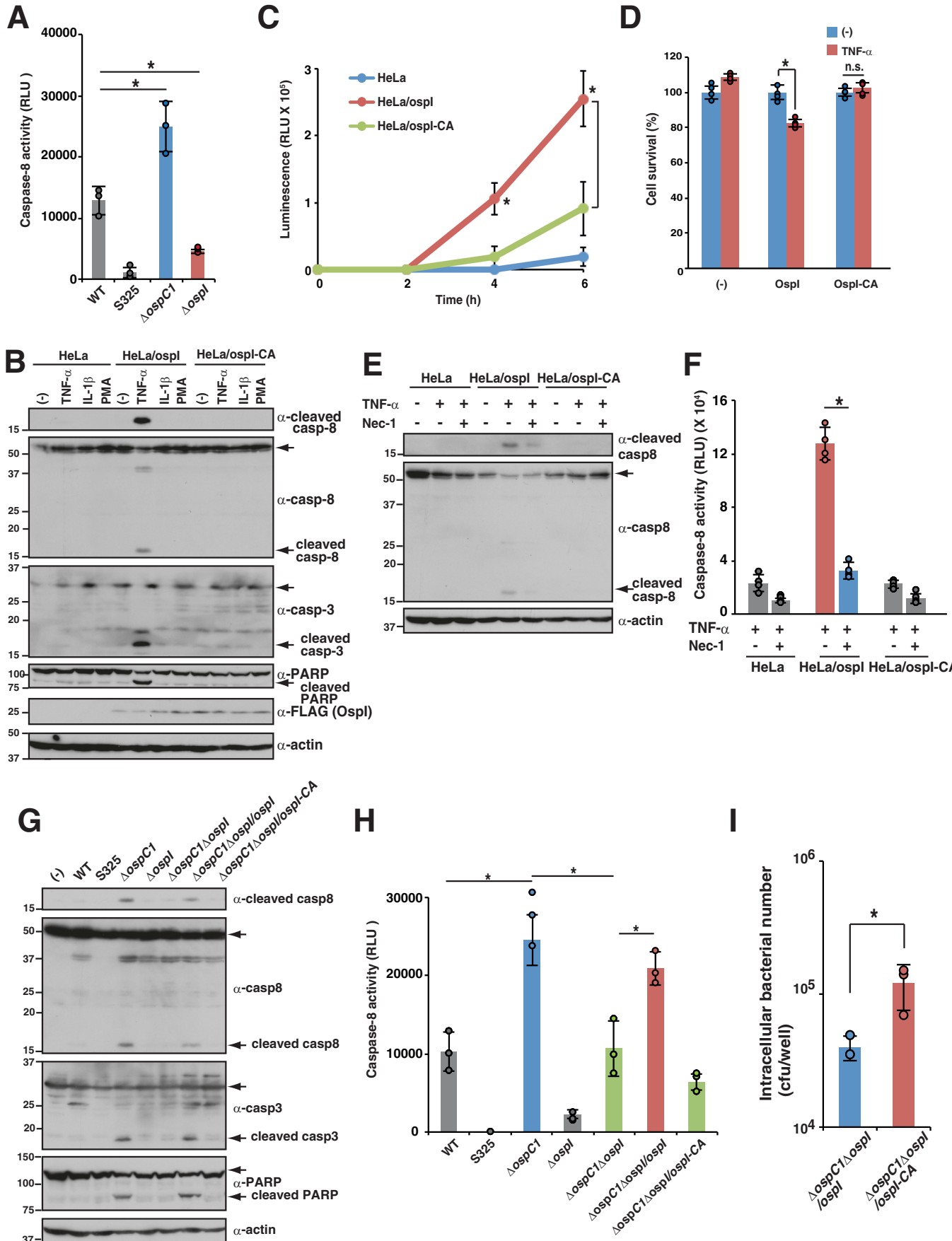

**Figure 2. The *Shigella* effector OspI triggers caspase-8 activation.**

(A) HT-29 cells were infected with the indicated *Shigella* strains and incubated for 8 h. Caspase-8 activity was measured and is reported as relative light units (RLU) of infected samples normalized to the value in uninfected samples. Data are expressed as the mean ± SD from triplicate and representative of three independent experiments ($P = 0.0039$ for $\Delta ospC1$ and $P = 0.025$ for $\Delta ospI$; one-way ANOVA). (B) HeLa cells stably expressing OspI or OspI-C62A were stimulated with TNF-α (25 ng/mL), IL-1β (50 ng/mL), or PMA (50 μg/mL) and incubated for 8 h. Cell lysates were subjected to immunoblotting. (C) HeLa cells stably expressing OspI or OspI-C62A were stimulated with TNF-α (25 ng/mL). After incubation for the indicated times, cells were subjected to real-time Annexin V apoptosis assay. Annexin V binding is reported as relative light units (RLU) in stimulated samples normalized to the value in unstimulated samples. Data are expressed as the mean ± SD from quadruplicate and representative of three independent experiments ($P = 0.0113$ for 4 η and $P < 0.0001$ for 6 h; two-way ANOVA). (D) HeLa cells stably expressing OspI or OspI-C62A were stimulated with TNF-α (25 ng/mL) and incubated for 8 h. Cells were subjected to MTT assay. Data are expressed as the mean ± SD from quadruplicate and representative of three independent experiments ($P < 0.0001$ for OspI and $P = 0.8419$ for OspI-CA; two-way ANOVA). (E, F) HeLa cells stably expressing OspI or OspI-C62A were stimulated with TNF-α (25 ng/mL) in the presence or absence of Nec-1 (50 μM), and incubated for 8 h. Cell lysates were subjected to immunoblotting (E) or measurement of caspase-8 activity (F). Caspase-8 activity is reported as relative light units (RLU) of the stimulated samples normalized to the values in unstimulated samples. Data are expressed as the mean ± SD from quadruplicate and representative of three independent experiments ($P < 0.0001$ for HeLa/ospI; two-way ANOVA). (G, H) HT-29 cells were infected with the indicated *Shigella* strains and incubated for 8 h. Cell lysates were subjected to immunoblotting (G) or measurement of caspase-8 activity (H). Caspase-8 activity is reported as relative light units (RLU) of infected samples normalized to the value in uninfected samples. Data are expressed as the mean ± SD from triplicate and representative of three independent experiments ($P$ values from left to right: $P < 0.0001$, $P < 0.0001$, and $P = 0.005$; two-way ANOVA). (I) HT-29 cells were infected with *Shigella* $\Delta ospC1 \Delta ospI/ospI$ or $\Delta ospC1 \Delta ospI/ospI$-CA. Intracellular bacteria were quantified using a gentamicin protection assay. Data are expressed as the mean ± SD from triplicate and representative of three independent experiments ($P = 0.03745$; two-tailed Student's $t$ test). Data are representative of three independent experiments (B, E, G). Molecular weights in immunoblots are in kDa. Data are considered significant when $P < 0.05$, with *$P < 0.05$ or n.s., not significant (A, C, D, F, H, I). Source data are available online for this figure.

demonstrating that TNFR signaling is required for caspase-8 activation (Fig. 1B,C). Because HT-29 cells infected with *Shigella* WT or $\Delta ospC1$ produced TNF-α, TNF-α secreted in response to *Shigella* infection can activate TNFR–caspase-8 signaling (Fig. 1D). A previous study has shown that TNF-α triggers apoptosis of *Shigella*-infected cells and restricts *Shigella* colonization in a mouse model (Roncaioli et al, 2023).

Activation of TNFR1 promotes RIPK1-dependent or RIPK1-independent caspase-8-mediated apoptosis. For example, blocking the translation downstream of NF-κB using cycloheximide (CHX) can induce RIPK1-independent caspase-8 activation. On the other hand, dysregulation of RIPK1 ubiquitination or phosphorylation using the inhibitor of apoptosis (IAP) antagonist, such as BV-6, can induce RIPK1-dependent caspase-8 activation (Wang et al, 2008). To determine whether $\Delta ospC1$-induced caspase-8 activation is RIPK1-dependent, we treated cells with Nec-1, an inhibitor of RIPK1 kinase activity. As shown in Fig. 1E,F, treatment with Nec-1 prevented caspase-8 cleavage or caspase-8 activity in $\Delta ospC1$-infected cells, suggesting that RIPK1-dependent caspase-8 activation is induced in $\Delta ospC1$-infected cells.

TNFR1 promotes the formation of a multiprotein complex comprising TRADD, FADD, and caspase-8 to initiate apoptosis. To further characterize the caspase-8 activation signaling pathway, we treated cells with the TRADD inhibitor apopstatin-1 (Apt-1) or ICCB-19 or siRNA targeting TRADD or FADD. Neither the TRADD inhibitor nor siRNA-mediated knockdown of TRADD or FADD in cells infected with $\Delta ospC1$ decreased the levels of cleaved caspase-8 (Fig. 1G,H). Collectively, these data indicate that the $\Delta ospC1$-induced caspase-8 activation pathway is different from the classical caspase-8–apoptosis pathway.

## The *Shigella* OspI effector triggers caspase-8 activation via TNFR signaling

Many bacterial pathogens deliver effectors to manipulate host cellular functions and prevent host innate immune responses. However, recent studies suggest that the host recognizes signaling disturbances by bacterial effectors and triggers alternative host defense systems in a process called effector-triggered immunity (Fischer et al, 2020; Remick et al, 2023). Because $\Delta ospC1$ induced a non-classical caspase-8 activation pathway, we hypothesized that signaling disturbance by *Shigella* T3SS effectors may become a cue to trigger caspase-8 activation.

To test our hypothesis, we searched for *Shigella* T3SS effectors that trigger caspase-8 activation. We infected HT-29 cells with various effector gene deletion mutants and measured caspase-8 activity (Appendix Fig. S1). Consistent with our previous report, cells infected with the $\Delta ospC1$ mutant showed significantly higher caspase-8 activity than those infected with WT *Shigella* (Fig. 2A). By contrast, cells infected with $\Delta ospI$ showed significantly lower caspase-8 activity than those infected with WT *Shigella* infection, suggesting that the OspI effector is required for caspase-8 activation (Fig. 2A).

OspI has deamidase activity, which prevents host inflammation by inhibiting NF-κB and mitogen-activated protein kinase (MAPK) activation (Sanada et al, 2012). When HeLa cells stably expressing OspI (HeLa/OspI) or deamidase activity-deficient OspI-C62A (HeLa/OspI-CA) were stimulated with TNF-α, IL-1β, or phorbol 12-myristate 13-acetate (PMA), HeLa/OspI cells exhibited caspase-8, caspase-3, and PARP activation, which are hallmarks of apoptosis, under TNF-α stimulation, but not IL-1β or PMA (Figs. 2B and EV1A,B). By contrast, in HeLa/OspI-CA cells, caspase-8, caspase-3, and PARP activation did not occur even under TNF-α stimulation (Figs. 2B and EV1A,B). The time course study of HeLa/OspI cells stimulated with TNF-α showed that cleaved caspase-8, caspase-3, and PARP began to accumulate at 2 h post-stimulation and resulted in inducing apoptosis and reducing cell viability (Figs. 2C,D and EV1C). Treatment with Nec-1 decreased the levels of cleaved caspase-8 in HeLa/OspI cells stimulated with TNF-α, indicating that OspI triggers RIPK1-dependent caspase-8 activation (Fig. 2E,F). Consistent with previous findings, OspI suppressed PMA-stimulated, but not TNF-α–stimulated, expression of the NF-κB-regulated gene *IL8* (Sanada et al, 2012) (Fig. EV1D). Although blocking translation downstream of NF-κB can induce RIPK1-independent caspase-8 activation, these data suggest that OspI triggers caspase-8 activation in NF-κB-independent manner.

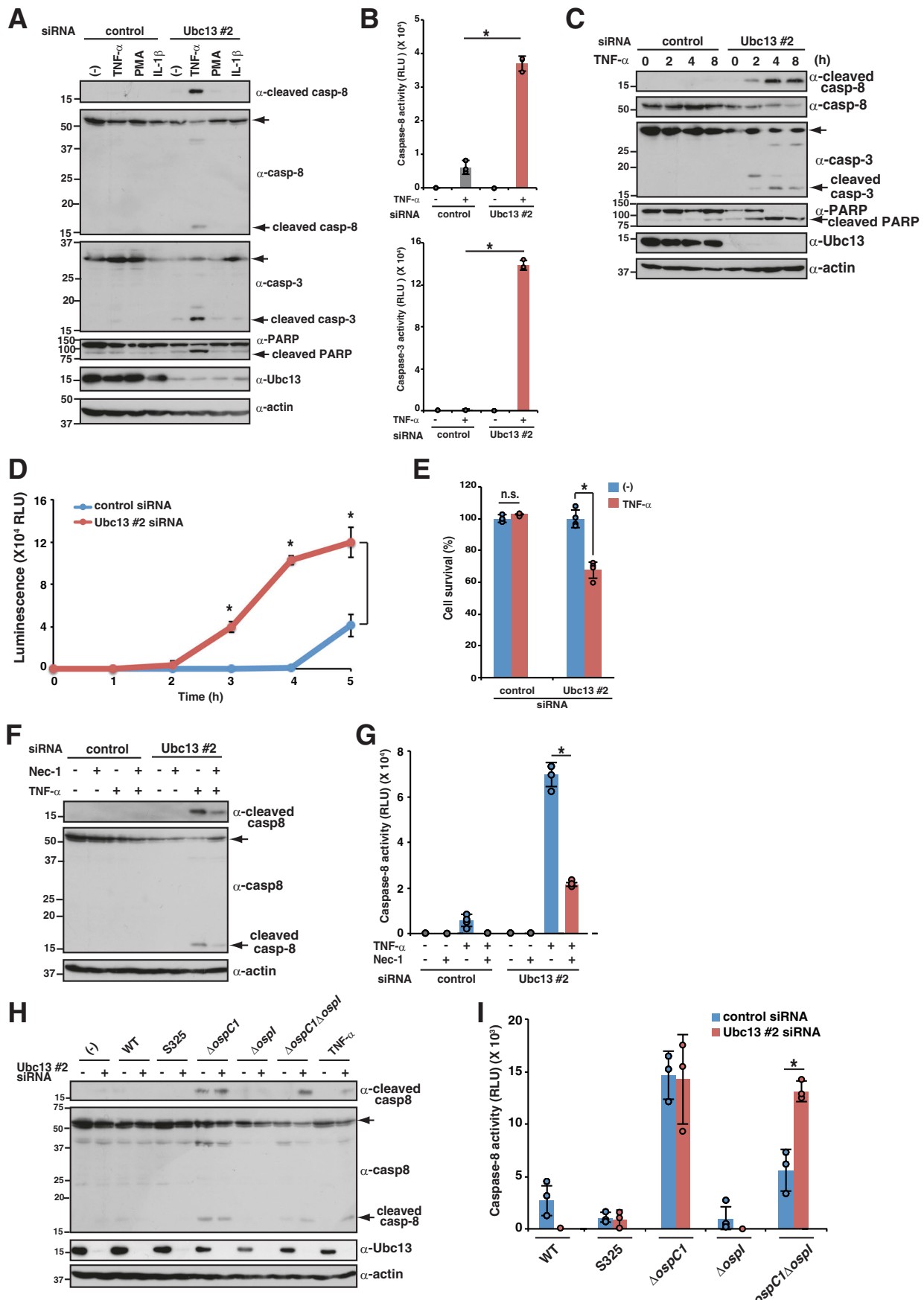

**Figure 3.** *Shigella* effector OspI-mediated Ubc13 inactivation triggers caspase-8 activation.

(A) HeLa cells treated with control or Ubc13 siRNAs were stimulated with TNF-α (25 ng/mL), IL-1β (50 ng/mL), or PMA (50 μg/mL) and incubated for 8 h. Cell lysates were subjected to immunoblotting. (B) HeLa cells treated with control or Ubc13 siRNAs were stimulated with TNF-α (25 ng/mL) and incubated for 8 h. Caspase-8 or caspase-3 activity was measured and is reported as relative light units (RLU) of stimulated samples normalized to the value in unstimulated samples. Data are expressed as the mean ± SD from triplicate and representative of three independent experiments ($P$ values: $P < 0.0001$ for caspase-8 and $P < 0.0001$ for caspase-3; two-way ANOVA). (C) HeLa cells treated with control or Ubc13 siRNAs were stimulated with TNF-α (25 ng/mL). At the indicated time points, cells were harvested and cell lysates were subjected to immunoblotting. (D) HeLa cells treated with control or Ubc13 siRNAs were stimulated with TNF-α (25 ng/mL) and incubated for up to 5 h. After incubation for the indicated times, cells were subjected to real-time Annexin V apoptosis assay. Annexin V binding is reported as relative light units (RLU) in stimulated samples, normalized to the value in unstimulated samples. Data are expressed as the mean ± SD from quadruplicate and representative of three independent experiments ($P$ values from left to right: $P < 0.0001$ for 3 h, $P < 0.0001$ for 4 h, $P < 0.0001$ for and 5 h; two-way ANOVA). (E) HeLa cells treated with control or Ubc13 siRNAs were stimulated with TNF-α (25 ng/mL) and incubated for 8 h. Cells were subjected to MTT assay. Data are expressed as the mean ± SD from quadruplicate and representative of three independent experiments ($P$ values: $P = 0.7481$ for control and $P < 0.0001$ for Ubc13 siRNA; two-way ANOVA). (F, G) HeLa cells treated with control or Ubc13 siRNA were stimulated with TNF-α (25 ng/mL) in the presence or absence of Nec-1 (50 μM) and incubated for 8 h. Cell lysates were subjected to immunoblotting (F) or measurement of caspase-8 activity (G). Caspase-8 activity is reported as RLU of stimulated samples normalized to the values in unstimulated samples. Data are expressed as the mean ± SD from quadruplicate and representative of three independent experiments ($P$ value < 0.0001; two-way ANOVA). (H) HeLa cells treated with control or Ubc13 siRNAs were infected with the indicated *Shigella* strains and incubated for 8 h. Cell lysates were subjected to immunoblotting. (I) HeLa cells treated with control or Ubc13 siRNAs were infected with the indicated *Shigella* strains and incubated for 8 h. Caspase-8 activity was measured and is reported as RLUs of infected samples normalized to the value in uninfected samples. Data are expressed as the mean ± SD from triplicate and representative of three independent experiments ($P = 0.0024$ for Δ*ospC1*Δ*ospI*; two-way ANOVA). Data are representative of three independent experiments (A, C, F, H). Molecular weights in immunoblots are in kDa. Data are considered significant when $P < 0.05$, with *$P < 0.05$ or n.s., not significant (B, D, E, G, I). Source data are available online for this figure.

Because the OspI effector leads to caspase-8 activation and apoptosis, we further investigated the correlation between OspI-mediated caspase-8 activation and OspC1-mediated caspase-8 inhibition. To this end, we measured the levels of caspase-8 activation in HT-29, HeLa, HCT116, or T84 cells infected with *Shigella* WT, S325 (T3SS-deficient mutant), Δ*ospC1*, Δ*ospI*, Δ*ospC1*Δ*ospI*, Δ*ospC1*Δ*ospI*/*ospI* (Δ*ospC1*Δ*ospI* complemented with *ospI*), or Δ*ospC1*Δ*ospI*/*ospI-CA* (Δ*ospC1*Δ*ospI* complemented with *ospI-CA mutant*). The results showed that the levels of cleaved caspase-8, caspase-3, and PARP were higher in cells infected with Δ*ospC1* than in cells infected with WT *Shigella*. In contrast to infection with Δ*ospC1*, infection with Δ*ospC1*Δ*ospI* did not lead to caspase-8, caspase-3, or PARP activation (Figs. 2G,H and EV2A–C). The reduction in the levels of cleaved caspase-8, caspase-3, and PARP in cells infected with Δ*ospC1*Δ*ospI* was rescued by *ospI* gene complementation (Δ*ospC1*Δ*ospI*/*ospI*), but not by *ospI-CA* gene complementation (Δ*ospC1*Δ*ospI*/*ospI-CA*), indicating that the deamidase activity of OspI results in caspase-8-mediated apoptosis during *Shigella* infection (Figs. 2G,H and EV2A–C). The fact that TNF-α stimulation failed to restore the levels of cleaved caspase-8 in Δ*ospC1*Δ*ospI*-infected cells further supported that functional OspI is required for caspase-8 activation during *Shigella* infection (Fig. EV2D). Consequently, the intracellular bacterial number of Δ*ospC1*Δ*ospI*/*ospI* was significantly lower than that of Δ*ospC1*Δ*ospI*/ *ospI-CA*, confirming that OspI-induced caspase-8 apoptosis acts as host defense to eliminate bacterial pathogens (Fig. 2I).

The time course of NF-κB and MAPK activation in Δ*ospI*-infected cells revealed increased phosphorylation of inhibitor of NF-κB (IκBα), degradation of IκBα, and phosphorylation of JNK as early as 40 min after infection in Δ*ospI*-infected cells relative to WT-infected cells, resulting in increasing inflammatory cytokine and chemokine production (Fig. EV2E,F). By contrast, a time course of caspase-8 activation in Δ*ospC1*-infected cells showed that cleaved caspase-8 began to accumulate at 6 h post-infection (Fig. EV2F). These data indicate the order of sequential T3SS effector function; OspI acts in the early stage of infection, whereas OspC1 acts in the late stage of infection, suggesting that OspI function is a cue to trigger caspase-8-mediated apoptosis. Taken together, these results indicate that the deamidase activity of OspI prevents NF-κB and MAPK activation, but also triggers caspase-8-mediated apoptosis as a trade-off, which is eventually prevented by OspC1.

## *Shigella* effector OspI-mediated Ubc13 inactivation triggers caspase-8 activation

OspI selectively targets and inactivates the E2 ubiquitin-conjugating activity of Ubc13, which is required for K63-linked polyubiquitination (Sanada et al, 2012). Because K63-linked polyubiquitin chains are involved in signal transduction cascades, including the NF-κB pathway, *Shigella* OspI prevents acute NF-κB–mediated inflammatory responses in the early stage of infection (Ashida et al, 2014; Madiraju et al, 2022). We therefore examined the effect of Ubc13 on OspI-dependent caspase-8 activation. Because the knockdown efficiency of Ubc13-targeting siRNAs was low in HT-29 cells, we performed Ubc13 knockdown experiments in HeLa or HCT116 cells. Consistent with the results in cells stably expressing OspI shown in Fig. 2, siRNA-mediated knockdown of Ubc13 increased caspase-8, caspase-3, and PARP activation in response to TNF-α stimulation, but not in response to IL-1β or PMA (Figs. 3A,B and E-V3A,B). A time course study of Ubc13-knockdown cells stimulated with TNF-α showed that cleavage of caspase-8, caspase-3, and PARP began to accumulate at 2 h post-stimulation and resulted in inducing apoptosis and reducing cell viability (Fig. 3C–E). To further characterize the caspase-8 activation pathway, Ubc13-knockdown cells were treated with Nec-1 and stimulated with TNF-α. In line with the data in Fig. 2E,F, treatment with Nec-1 prevented cleavage of caspase-8 in Ubc13-knockdown cells, indicating that Ubc13 knockdown induced RIPK1-dependent caspase-8 activation (Fig. 3F,G). Consistent with a previous study, Ubc13 knockdown suppressed PMA-stimulated, but not TNF-α-stimulated, NF-κB-regulated *IL8* expression, suggesting that inactivation of Ubc13 triggers caspase-8 activation in an NF-κB-independent manner (Fig. EV3C) (Xu et al, 2009). Collectively, these findings indicate that OspI-mediated Ubc13 inactivation triggers caspase-8-mediated apoptosis via TNFR signaling.

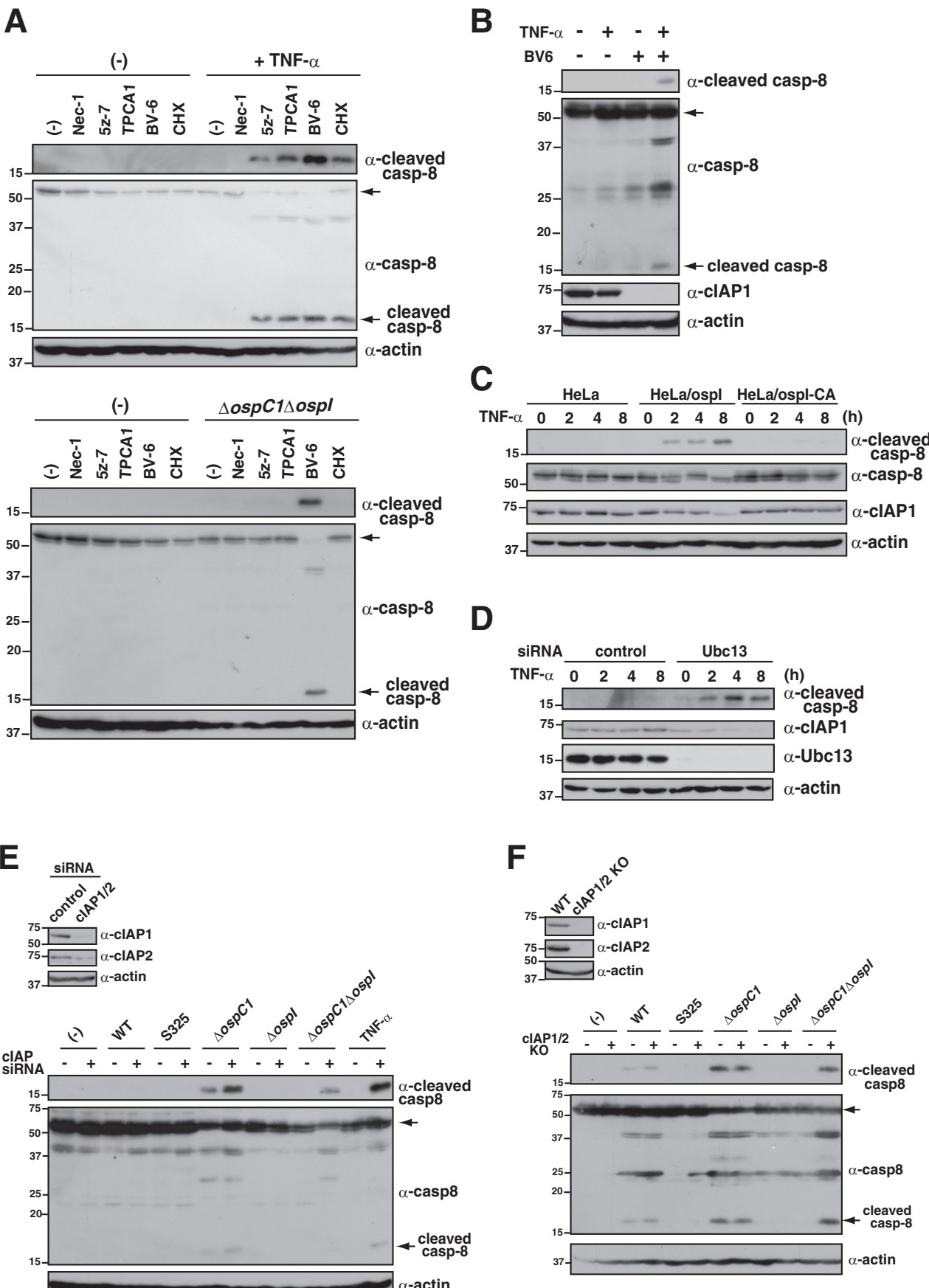

**Figure 4.  *Shigella* effector OspI-mediated cIAP inactivation triggers caspase-8 activation.**

(A) HeLa cells were stimulated with TNF-α (25 ng/mL) or infected with *Shigella ΔospC1ΔospI* in the presence or absence of Nec-1 (50 μM), 5z-7 (0.5 μM), TPCA1 (5 μM), BV-6 (1 μM), or CHX (25 μg/mL) and incubated for 8 h. Cell lysates were subjected to immunoblotting. (B) HeLa cells were stimulated with TNF-α (25 ng/mL) in the presence or absence of BV-6 (1 μM) and incubated for 8 h. Cell lysates were subjected to immunoblotting. (C) HeLa cells stably expressing OspI or OspI-C62A were stimulated with TNF-α (25 ng/mL). At the indicated time points, cells were harvested and cell lysates were subjected to immunoblotting. (D) HeLa cells treated with control or Ubc13 siRNAs were stimulated with TNF-α (25 ng/mL). At the indicated time points, cells were harvested and cell lysates were subjected to immunoblotting. (E) HeLa cells treated with control or cIAP1 plus cIAP2 siRNAs were infected with the indicated *Shigella* strains and incubated for 8 h. Cell lysates were subjected to immunoblotting. The knockdown efficiency of the indicated siRNAs was assessed by immunoblotting (inset). (F) HT-29 WT or cIAP1/2 KO cells were infected with the indicated *Shigella* strains and incubated for 8 h. Cell lysates were then subjected to immunoblotting. The knockout of the indicated genes was assessed by immunoblotting (inset). All data are representative of three independent experiments. Molecular weights in immunoblots are in kDa. Source data are available online for this figure.

To confirm that inactivation of Ubc13 triggers caspase-8 activation during *Shigella* infection, cells transfected with siRNA targeting Ubc13 were infected with a series of *Shigella* gene deletion mutants (Figs. 3H,I and EV3D,E). The increase in cleaved caspase-8 observed in cells infected with *ΔospC1* was abolished in cells infected with *ΔospC1ΔospI*; however, knockdown of Ubc13 rescued the cleavage of caspase-8 in cells infected with *ΔospC1ΔospI* (Figs. 3H,I and EV3D,E). Taken together, these findings indicate that OspI-mediated Ubc13 inactivation triggers caspase-8 activation under TNF-α stimulation. To counteract this, *Shigella* delivers OspC1 and prevents OspI-induced caspase-8-mediated apoptosis during *Shigella* infection.

## *Shigella* effector OspI-mediated cIAP inactivation triggers caspase-8 activation

K63-linked ubiquitination by Ubc13 affects several signaling pathways, including NF-κB and MAPK (Ashida et al, 2014; Madiraju et al, 2022). We therefore investigated which host factor is targeted by OspI-Ubc13, resulting in the activation of caspase-8. To this end, HeLa cells or HT-29 cells infected with *Shigella ΔospC1ΔospI*, which is deficient in caspase-8 activation, were treated with several inhibitors, including Nec-1 (RIPK1 inhibitor), 5z-7 (TAK1 inhibitor), TPCA-1 (IKK-β inhibitor), BV-6 (IAP inhibitor), and CHX (protein translation inhibitor). Although the increase in caspase-8 activation observed in cells infected with *ΔospC1* was abolished in cells infected with *ΔospC1ΔospI*, only BV-6 treatment rescued caspase-8 activation in cells infected with *ΔospC1ΔospI* (Figs. 4A and EV4A,B). These results indicate that the OspI-Ubc13 axis might target IAP and trigger caspase-8 activation during *Shigella* infection. On the other hand, TPCA-1, CHX or BAY11-7082 (NF-κB inhibitor) treatment did not rescue caspase-8 activation in cells infected with *ΔospC1ΔospI*, suggesting that NF-κB-regulated translation of anti-apoptotic proteins, including cFLIP, is not important for *Shigella*-induced caspase-8 activation (Figs. 4A and EV4A–C) (Wang et al, 2008). Furthermore, because the loss of another NF-κB-inhibiting effector—OspG, which binds to E2 enzyme UbcH5b to downregulate NF-κB activation—did not affect caspase-8 activity, the OspI-Ubc13 axis appears to be specific for caspase-8 activation (Fig. EV4D,E) (Kim et al, 2005; Pruneda et al, 2014).

cIAP1, cIAP2, and XIAP are members of the IAP family of apoptosis inhibitors (Estornes and Bertrand, 2015; Peltzer et al, 2016). cIAP1 and cIAP2 have E3 ligase activity mediating both K63-linked and K48-linked polyubiquitination (Estornes and Bertrand, 2015). In response to TNF-α stimulation, cIAP1/2 catalyzes K63-linked polyubiquitination of RIPK1, which recruits

the TAK1/TAB complex to induce the activation of NF-κB. The polyubiquitination of RIPK1 catalyzed by cIAP1/2 in TNF-α-stimulated cells also prevents the formation of a protein complex that would activate caspase-8 (Wang et al, 2008; Vince et al, 2007; Bertrand et al, 2008; Mahoney et al, 2008; Zhang et al, 2024). Therefore, the absence or inactivation of cIAP1/2 promotes caspase-8 activation and induces apoptosis. Because BV-6 binds to IAPs and leads to their degradation, cells treated with TNF-α plus BV-6 showed activation of caspase-8 and decreased levels of cIAP1 (Fig. 4B). When cells were treated with TNF-α, the levels of cIAP1 decreased gradually in cells stably expressing OspI, but not in cells stably expressing OspI-CA (Fig. 4C). Furthermore, when Ubc13-knockdown cells were treated with TNF-α, the levels of cIAP1 decreased gradually in Ubc13-knockdown cells (Fig. 4D). These data suggest that decreased levels of cIAP by OspI-Ubc13 correlates with caspase-8 activation.

To support our hypothesis, cells treated with siRNA targeting cIAP1/2 or cIAP1/2-double-knockout cells were infected with a series of *Shigella* gene deletion mutants. Although the increase in cleaved caspase-8 observed in cells infected with *ΔospC1* was abolished in cells infected with *ΔospC1ΔospI*, knockdown or knockout of cIAP1/2 rescued the cleavage of caspase-8 in cells infected with *ΔospC1ΔospI* (Fig. 4E,F).

The results suggest that OspI-mediated Ubc13 inactivation triggers cIAP inactivation, resulting in activation of caspase-8-mediated apoptosis, which is eventually prevented by OspC1, during *Shigella* infection.

## The *Shigella* OspC1 effector prevents caspase-8 activation via its ADP-riboxanation activity

Many T3SS effectors have specific enzymatic activities that mimic and usurp host cellular function. In a previous study, we demonstrated that the *Shigella* OspC1 effector blocks caspase-8 activation to prevent apoptosis. However, the phenotypes of OspC1 and the precise mechanism of caspase-8 inhibition by OspC1 remain unclear (Ashida et al, 2020). Several recent studies showed that *Shigella* OspC family and CopC, an OspC3 homolog in *Chromobacterium violaceum*, possess arginine ADP-riboxanase enzymatic activity (Li et al, 2021; Peng et al, 2022; Liu et al, 2022; Zhang et al, 2022; Hou et al, 2023).

Therefore, to gain insight into the mechanism by which OspC1 inhibits caspase-8, we investigated whether OspC1 catalyzes the ADP-riboxanation of caspase-8. We constructed mutants in which the aspartic acid residue at position 172 or 226 was replaced by alanine and examined their effects on caspase-8 inhibition (Fig. 5A). Given that OspC3 and CopC transfer ADP-ribose from

## A

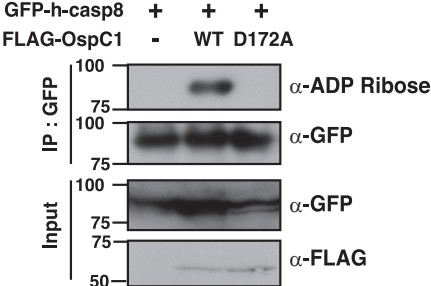

OspC1 165 KRNTYAD**D**IKKIANHDFVFFGVEISNHQKKHPLNTKHHTV 204
OspC2 170 YKNTFSE**D**IEEIANHDFVFFGVEISNHQETLPLNKTHHTV 209
OspC3 170 YKNTFSE**D**IEEIANHDFVFFGVEISNHQETLPLNKTHHTV 209

OspC1 205 DFGANAYIIDHDSPYGYMTLT**D**HFDNAIPPVFYHEHQS–F 243
OspC2 210 DFGANAYIIDHDSPYGYMTLT**D**HFDNAIPPVFYHEHQSFF 249
OspC3 210 DFGANAYIIDHDSPYGYMTLT**D**HFDNAIPPVFYHEHQSFF 249

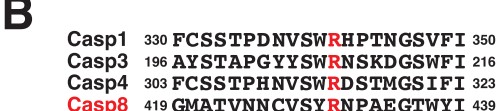

## B

Casp1 330 FCSSTPDNVSW**R**HPTNGSVFI 350
Casp3 196 AYSTAPGYYSW**R**NSKDGSWFI 216
Casp4 303 FCSSTPHNVSW**R**DSTMGSIFI 323
Casp8 419 GMATVNNCVSY**R**NPAEGTWYI 439

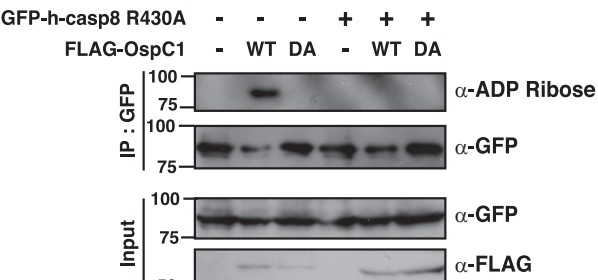

## C

h-Casp8 419 GMATVNNCVSY**R**NPAEGTWYI 439
m-Casp8 404 GMATVKNCVSY**R**DPVNGTWYI 424

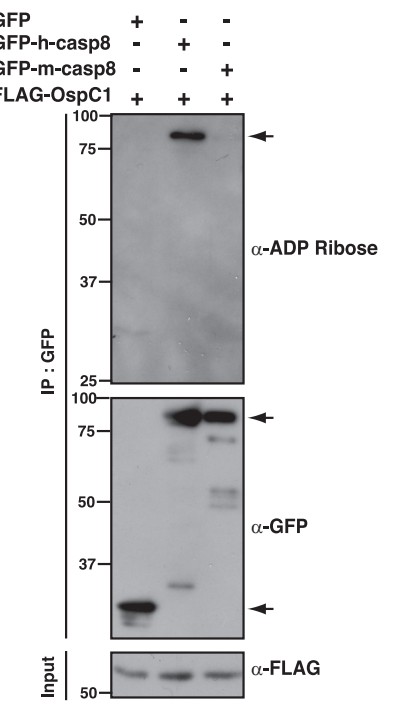

## D

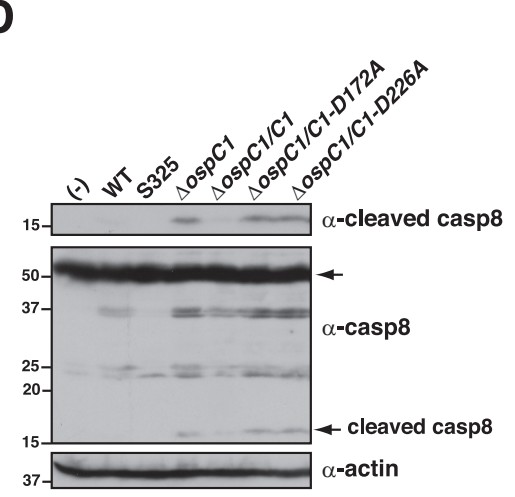

## E

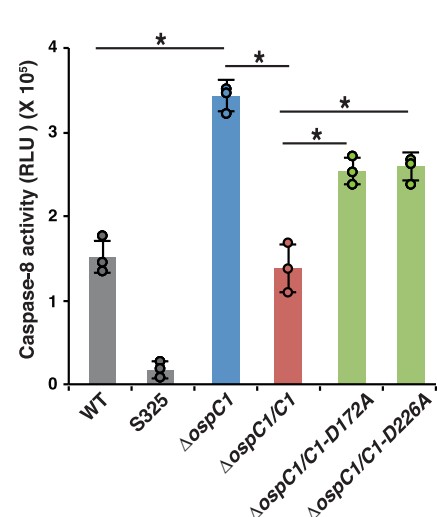

◀ **Figure 5. The ADP-riboxanation activity of OspC1 is required for caspase-8 inhibition.**

(A) Multiple sequence alignments of the *Shigella* OspC family OspC1, OspC2, and OspC3. Conserved amino acids involved in arginine ADP-riboxanation activity are colored in red (top). 293T cells were transfected with the indicated plasmids. After 24 h, cells were harvested and subjected to immunoprecipitation and immunoblotting (bottom). (B) Alignments of caspase sequences. Conserved arginine residues are colored in red (top). 293T cells were transfected with the indicated plasmids. After 24 h, cells were harvested and subjected to immunoprecipitation and immunoblotting (bottom). (C) Alignments of human and mouse caspase-8 sequences (top). The 293T cells were transfected with the indicated plasmids. After 24 h, cells were harvested and subjected to immunoprecipitation and immunoblotting (bottom). (D) HT-29 cells were infected with the indicated *Shigella* strains and incubated for 8 h. Cell lysates were subjected to immunoblotting. (E) HT-29 cells were infected with the indicated *Shigella* strains and incubated for 8 h. Cells were harvested and caspase-8 activity was measured. Caspase-8 activity is reported as relative light units (RLU) of infected samples normalized to the value in uninfected samples. Data are expressed as the mean ± SD from triplicate and representative of three independent experiments (*P* values from left to right: $P < 0.0001$, $P < 0.0001$, $P < 0.0001$, and $P < 0.0001$; two-way ANOVA). Data are representative of three independent experiments (A–D). Molecular weights in immunoblots are in kDa. Data are considered significant when $P < 0.05$, with *$P < 0.05$ (E). Source data are available online for this figure.

nicotinamide adenine dinucleotide (NAD⁺) to the arginine residue of target proteins, we co-transfected caspase-8 with an OspC1– or OspC1-D172A–expressing plasmid in 293 T cells and detected protein modification by immunoblotting (Li et al, 2021; Peng et al, 2022). The results indicated that OspC1, but not OspC1-D172A, catalyzed the ADP-riboxanation of caspase-8, suggesting that OspC1 directly inhibits caspase-8 (Fig. 5A). Activation of caspase-8 leads to the release of the active subunits p18 and p10 and the activation of downstream effector caspases such as caspase-3 and -7. We next investigated which arginine residue of caspase-8 is targeted by OspC1. The p10 subunit of caspase-8 contains Arg430, which is a conserved residue in caspases (Fig. 5B). Because the same conserved Arg residue in caspase-4 or caspase-3 is targeted by OspC3 or CopC, respectively, we hypothesized that OspC1 might target and catalyze the ADP-riboxanation of Arg430 in caspase-8 (Li et al, 2021; Peng et al, 2022). As shown in Fig. 5B, OspC1 catalyzed the ADP-riboxanation of WT caspase-8, but not that of the R430A caspase-8 mutant, suggesting that OspC1 modifies Arg430 of caspase-8. Because some bacterial effectors are species-specific (Luchetti et al, 2021), we further analyzed OspC1-catalyzed ADP-riboxanation using human and mouse caspase-8. Although Arg415 (corresponding to Arg430 in human caspase-8) is conserved in mouse caspase-8, OspC1 could not catalyze the ADP-riboxanation of mouse caspase-8 (Fig. 5C). These data may explain the species-specific difference for *Shigella*. To further determine the functional importance of ADP-riboxanase activity in OspC1, we measured the levels of cleaved caspase-8 and caspase-8 activity in HT-29, HeLa, HCT116 and T84 cells infected with *Shigella* WT, S325, *ΔospC1*, *ΔospC1/C1* (*ΔospC1* complemented with *ospC1*), *ΔospC1/C1-D172A* (*ΔospC1* complemented with *ospC1-D172A mutant*), or *ΔospC1/C1-D226 A* (*ΔospC1* complemented with *ospC1-D226A mutant*). As shown in Figs. 5D,E and EV5, the levels of cleaved caspase-8 and caspase-8 activity were reduced in cells infected with *Shigella* WT or *ΔospC1/C1*, but not *ΔospC1*, *ΔospC1/C1-D172A*, or *ΔospC1/C1-D226A*, indicating that the ADP-riboxanation activity of OspC1 is important for preventing caspase-8 activation. These results indicate that OspC1 directly inhibits caspase-8 activation via its ADP-riboxanase activity.

In addition to the inhibition of cell death by OspC1, recent studies identified a novel effector function of the *Shigella* OspC family. Zhang et al found that *Shigella* OspC family effectors induce stress granule formation, which inhibits host protein synthesis and promotes bacterial replication during infection in an ADP-riboxanase activity-dependent manner (Zhang et al, 2024). On the other hand, Alphonse et al demonstrated that OspC1 and OspC3 block interferon signaling and attenuate antibacterial

responses by inhibiting CaMKII and STAT1 phosphorylation, which is mediated by binding to CaM in an ADP-riboxanase activity-independent manner (Alphonse et al, 2022). These findings suggest that OspC1 acts as a multifunctional effector that prevents both inflammation and cell death during *Shigella* infection.

## *Shigella* effector OspI-mediated caspase-8 activation and OspC1-mediated caspase-8 inhibition are both required for necroptosis induction

Caspase-8 is the molecular switch for apoptosis, necroptosis, and pyroptosis. Inhibition or loss of caspase-8 in the apoptotic pathway leads to necroptotic cell death (Günther et al, 2011). Caspase-8 triggers apoptosis by activating caspase-3, but it also cleaves RIPK1 and RIPK3, thereby inhibiting necroptosis. Inactivation or inhibition of caspase-8 induces phosphorylation of RIPK1 and RIPK3, resulting in the induction of necroptosis through the activation of MLKL (Pasparakis and Vandenabeele, 2015). In this study, TNF-α + BV-6 (RIPK-dependent) and TNF-α + CHX (RIPK-independent) treatments both activated caspase-8, but not phosphorylation of RIPK1 and RIPK3 (Fig. EV6A). Inhibition of caspase-8 by z-VAD treatment in the presence of these stimulants resulted in the phosphorylation of RIPK1, RIPK3, and MLKL, and increased cytotoxicity, which are hallmarks of necroptosis (Fig. EV6A,B).

We previously demonstrated that *Shigella* delivers the OspD3 effector to prevent necroptosis (Ashida et al, 2020). The number of intracellular bacteria of the *ΔospD3* mutant was significantly lower than that of the WT, indicating that prevention of necroptosis by OspD3 contributes to prolonged bacterial colonization (Fig. 6A). Therefore, we next investigated the correlation between caspase-8 activation by OspI, caspase-8 inhibition by OspC1, and necroptosis inhibition by OspD3. To this end, we first investigated the correlation between caspase-8 activation by OspI and necroptosis inhibition by OspD3. We measured the levels of phosphorylated MLKL, RIPK1, and RIPK3, and cytotoxicity, which are hallmarks of necroptosis, in HT-29 and HT-55 cells infected with series of *Shigella* effector gene deletion mutants (Figs. 6B,C and EV6C). The results showed that the levels of phosphorylated MLKL, RIPK1, and RIPK3, and cytotoxicity were higher in cells infected with *ΔospD3* than in cells infected with WT *Shigella*, indicating that *ΔospD3* infection triggers necroptosis (Fig. 6B,C). In contrast to infection with *ΔospD3*, infection with *ΔospD3ΔospI* did not increase the levels of phosphorylated MLKL, RIPK1, and RIPK3, and cytotoxicity (Fig. 6B,C). The reduction in the levels of phosphorylated MLKL, RIPK1, and RIPK3, and cytotoxicity in cells infected with *ΔospD3ΔospI* was rescued by *ospI* gene complementation, but not

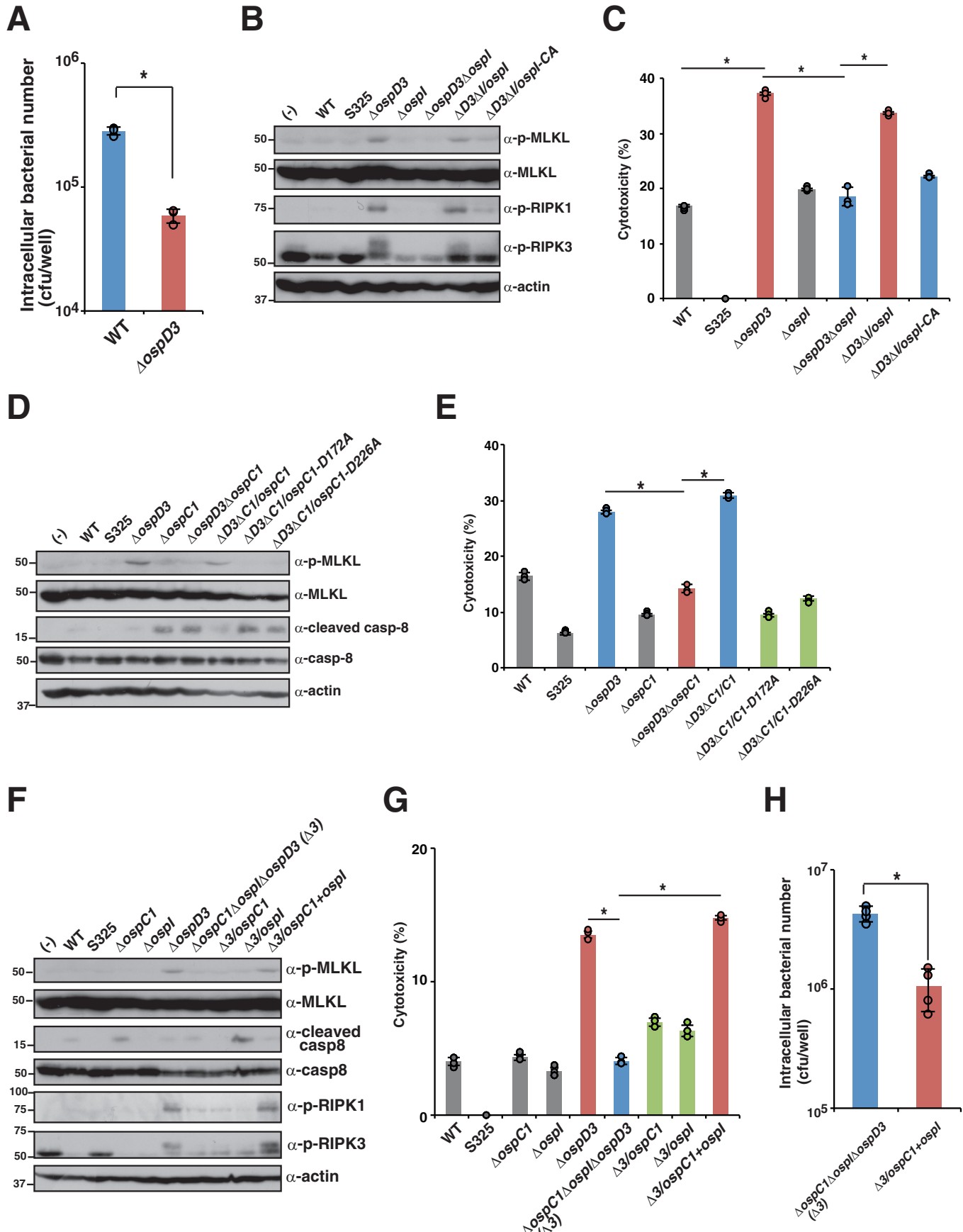

◀

**Figure 6. Both *Shigella* effector OspI-mediated caspase-8 activation, and OspC1-mediated caspase-8 inhibition are required for necroptosis.**

(A) HT-29 cells were infected with *Shigella* WT or *ΔospD3* and intracellular bacteria were quantified using a gentamicin protection assay. Data are expressed as the mean ± SD from triplicate and representative of three independent experiments (*P* = 0.000075; two-tailed Student's *t* test). (B–G) HT-29 cells were infected with the indicated *Shigella* strains and incubated for 8 h. Cell lysates and aliquots of cellular supernatants were subjected to immunoblotting (B, D, F) and cytotoxicity assays (C, E, G), respectively. (C) Data are expressed as the mean ± SD from triplicate and representative of three independent experiments (*P* values from left to right: *P* < 0.0001, *P* < 0.0001, and *P* < 0.0001; two-way ANOVA). (E) Data are expressed as the mean ± SD from triplicate and representative of three independent experiments (*P* values from left to right; *P* < 0.0001 and *P* < 0.0001; two-way ANOVA). (G) Data are expressed as the mean ± SD from triplicate and representative of three independent experiments (*P* values: *P* < 0.0001 (left) and *P* < 0.0001 (right); two-way ANOVA). (H) HT-29 cells were infected with *Shigella ΔospC1ΔospD3ΔospI* or *ΔospC1ΔospD3ΔospI/ospC1+ospI*. Intracellular bacteria were quantified using a gentamicin protection assay. Data are expressed as the mean ± SD from quadruplicate and representative of three independent experiments (*P* = 0.00013; two-tailed Student's *t*-test). Data are representative of three independent experiments (B, D, F). Molecular weights in immunoblots are in kDa. Data are considered significant when *P* < 0.05, with **P* < 0.05 (A, C, E, G, H). Source data are available online for this figure.

by *ospI-CA* gene complementation (Fig. 6B,C). These data suggest that OspI-mediated caspase-8 activation is required for the induction of necroptosis during *Shigella* infection.

In addition, caspase-8 inhibition by OspC1 via its ADP-riboxanation activity becomes a cue to trigger necroptosis (Fig. 6D,E). In contrast to infection with *ΔospD3*, infection with *ΔospC1ΔospD3* did not increase the levels of phosphorylation of MLKL or cytotoxicity, whereas caspase-8 activity was elevated in cells infected with *ΔospC1* or *ΔospC1ΔospD3* (Fig. 6D,E). The decrease in the levels of MLKL phosphorylation and cytotoxicity in cells infected with *ΔospC1ΔospD3* was rescued by *ospC1* gene complementation (*ΔospC1ΔospD3/ospC1*), but not by *ΔospC1ΔospD3/ospC1-D172A* or *ΔospC1ΔospD3/ospC1-D226A* (Fig. 6D,E). These results suggest that OspC1 inhibits caspase-8 activation via its ADP-riboxanase activity and induces necroptosis, which is eventually inhibited by OspD3.

Finally, we aimed to confirm the correlation between OspI, OspC1, and OspD3. In contrast to infection with *ΔospD3*, infection with *ΔospC1ΔospD3ΔospI* did not increase the levels of phosphorylated MLKL, RIPK1, and RIPK3, and cytotoxicity (Fig. 6F,G). The reduction in the levels of phosphorylated MLKL, RIPK1, and RIPK3, and cytotoxicity in cells infected with *ΔospC1ΔospD3ΔospI* was rescued by complementation with both *ospC1* and *ospI* genes (*ΔospC1ΔospD3ΔospI/ospC1+ospI*), but not with each *ospC1* or *ospI* single gene complementation (*ΔospC1ΔospD3ΔospI/ospC1* or *ΔospC1ΔospD3ΔospI/ospI*) (Fig. 6F,G). These data indicate that both OspI-mediated caspase-8 activation and OspC1-mediated caspase-8 inhibition are required for triggering necroptosis, which is eventually counteracted by the OpsD3 effector. Consequently, the intracellular bacterial number of *ΔospC1ΔospD3ΔospI/ospC1+ospI* was significantly lower than that of *ΔospC1ΔospD3ΔospI*, confirming that necroptosis induced by both OspI and OspC1 acts as host defense to eliminate bacterial pathogens (Fig. 6H). Overall, these data suggest that necroptosis is induced during *Shigella* infection via both activation of caspase-8 by OspI and inhibition of caspase-8 by OspC1; these processes are the mediators of a *Shigella* survival strategy that counteracts inflammation and apoptosis. However, to counteract this, *Shigella* delivers another effector, OspD3, and prevents necroptosis to promote bacterial infection (Fig. 7).

## Discussion

Host cells respond to bacterial infection by activating different cell death programs to expel infected cells, eliminate invasive pathogens, and activate innate and adaptive immune systems. Upon *Shigella* invasion and multiplication within host cells, PRRs, such as

nucleotide-binding oligomerization domain-like receptors (NLRs), detect PAMPs or DAMPs, and activate cell death signaling (Ashida et al, 2021). For example, the NAIPs, which are members of a subfamily of NLRs, detect *Shigella* T3SS components and activate the NLRC4-inflammasome and pyroptosis (Kofoed and Vance, 2011; Zhao et al, 2011; Rayamajhi et al, 2013; Yang et al, 2013; Suzuki et al, 2013). Indeed, guanylate-binding proteins (GBPs) act as cytosolic LPS sensors and form a complex that promotes caspase-4 activation, resulting in caspase-4-dependent pyroptosis (Shi et al, 2015, Wandel et al, 2020; Santos et al, 2020). On the other hand, the phagocytic membrane remnants or oxidative stress generated by *Shigella* infection are detected by host cells as DAMPs, triggering pyroptosis or necrosis-like cell death (Carneiro et al, 2009; Dupont et al, 2009). Intriguingly, PAMPs or DAMPs are not the only mechanisms of pathogen recognition and stimulation for triggering host cell death. Host cells also recognize the disruption of host cell signaling by T3SS effector activities, which triggers an alternative innate immune response termed "effector-triggered immunity" as a backup host defense system (Fischer et al, 2020; Remick et al, 2023).

In this study, we describe a mechanism of cell death induction known as effector-triggered immunity and the bacterial counter-measures during *Shigella* infection. We demonstrate that the stimulant that drives caspase-8-mediated apoptosis and necroptosis during *Shigella* infection is bacterial T3SS effector activity, but not PAMPs or DAMPs. We provide first evidence that the circumvention of host inflammatory signaling by the OspI effector becomes a cue to trigger apoptosis and necroptosis. The *Shigella* OspI effector inactivates Ubc13 via its deamidase activity and inhibits NF-κB activation, thereby downregulating host inflammatory responses (Sanada et al, 2012). We discovered that host cells identify the inactivation of Ubc13-cIAP by OspI and trigger caspase-8-mediated apoptosis or necroptosis via effector-triggered immunity (Fig. 7).

The RING-type E3 ligases cIAP1 and cIAP2 ubiquitinate RIPK1 and block its kinase activity, thereby inhibiting TNF-α–mediated apoptosis (Wang et al, 2008; Vince et al, 2007; Varfolomeev et al, 2007; Bertrand et al, 2008; Mahoney et al, 2008; Schorn et al, 2023; Zhang et al, 2024). Therefore, loss or inactivation of cIAPs induces overactivation of RIPK1 and eventually leads to caspase-8-mediated apoptosis. We found that inactivation of Ubc13 triggers degradation of cIAP1 under TNF-α stimulation (Fig. 4). A Recent study revealed that K63-linked polyubiquitination mediated by Ubc13 regulates self-ubiquitination and proteasomal degradation of cIAP1 (Akizuki et al, 2022). Because cIAPs catalyze the K63-linked polyubiquitination of RIPK1, which is required for preventing cell

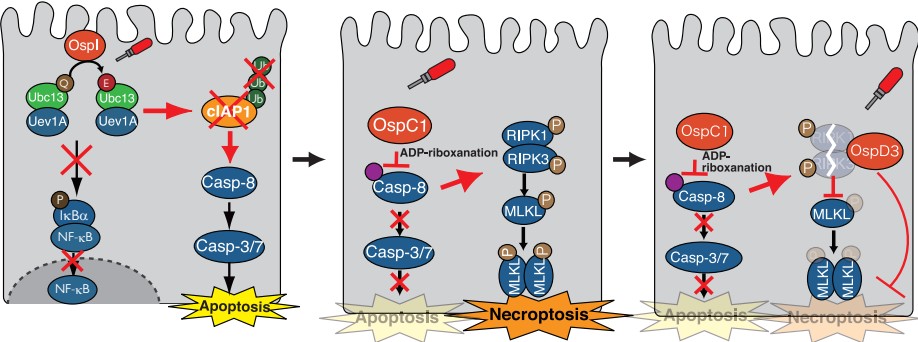

**Figure 7. Model of this study.**

(i) When *Shigella* invades and multiplies within epithelial cells, PAMPs and DAMPs are released. Host cells detect these PAMPs and DAMPs and activate NF-κB. *Shigella* delivers the OspI effector and deamidates the E2 ubiquitin conjugative enzyme Ubc13. OspI inactivates Ubc13, thereby inhibiting NF-κB activation but also inducing caspase-8-mediated apoptosis through cIAP inactivation. (ii) To counteract this, *Shigella* delivers the OspC1 effector, preventing caspase-8 activation and apoptotic cell death via its ADP-riboxanation activity. (iii) By contrast, host cells detect the bacteria-mediated disturbance of caspase-8 activation, resulting in the induction of necroptosis as a backup host defense. (iv) *Shigella* subsequently delivers the OspD3 effector, which targets RIPK1 and RIPK3 for degradation via its protease activity to prevent necroptosis.

death, inactivation of Ubc13 by OspI might affect the E3 ligase activity of cIAPs and its stability, resulting in RIPK1-dependent apoptosis (Peltzer et al, 2016; Vince et al, 2007; Bertrand et al, 2008; Zhang et al, 2024).

The results of this study showed that inactivation of Ubc13 by OspI specifically triggered TNF-α–, but not PMA-mediated caspase-8 activation, whereas inactivation of Ubc13 by OspI selectively inhibited PMA-, but not TNF-α-mediated NF-κB activation. PMA is a substitute for diacylglycerol, indicating that OspI modulates the diacylglycerol–CBM (CARMA-BCL10-MALT1) complex–TRAF6-NF-κB signaling pathway during *Shigella* infection (Sanada et al, 2012). In other words, *Shigella* OspI targets the Ubc13-TRAF6 complex and inhibits NF-κB activation at the expense of inducing caspase-8-mediated apoptosis. These data suggest that OspI acts as a double-edged sword, inhibiting NF-κB–dependent inflammatory responses, but ultimately inducing apoptosis and promoting bacterial elimination.

The results of this study suggest that *Shigella* evolved sophisticated strategies for evading or preventing effector-triggered immunity. *Shigella* delivers the OspC1 effector, which has ADP-riboxanase activity, and inhibits caspase-8 activation induced by OspI (Figs. 5 and 7). However, host cells recognize the blockade of caspase-8-mediated apoptosis signaling by the OspC1 effector and trigger necroptosis as secondary effector-triggered immunity (Ashida et al, 2020).

In addition to elucidating the function of OspC1, we demonstrated that OspI-induced caspase-8 activation mediated by the inactivation of cIAP becomes a cue to trigger necroptosis (Figs. 4 and 7). Loss of cIAP sensitizes cells to apoptosis, whereas inhibition of caspase-8 sensitizes cells to necroptosis (Vince et al, 2007; Varfolomeev et al, 2007; He et al, 2009). Therefore, although host cells recognize the inactivation of cIAP by OspI and trigger cell death, whether cell death occurs via apoptosis or necroptosis depends on caspase-8 activity. Namely, OspC1 acts as a cell death gatekeeper that shifts OspI-induced apoptosis to necroptosis, which is eventually prevented by the OspD3 effector.

Although this study examined a cell death induction mechanism known as effector-triggered immunity and bacterial countermeasures during *Shigella* infection, future studies should conduct investigations using more native systems, such as an in vivo infection model or in primary cells. Although mice are naturally resistance to oral *Shigella* inoculation, the Vance group has developed NAIP-NLRC4 inflammasome-deficient mouse model as a useful tool for in vivo investigations of mouse shigellosis (Mitchell et al, 2020). However, our data demonstrated that OspC1 specifically catalyzed the ADP-riboxanation of human caspase-8 but not mouse caspase-8, suggesting that the NAIP-NLRC4 inflammasome-deficient mouse model is not suitable for our study (Fig. 5). We intend to address this issue using human organoids or primary cells in future studies, which would reveal the physiological relevance of our findings of effector-triggered immunity and bacterial countermeasures.

In conclusion, we elucidated the multilayered strategies used by *Shigella* to counteract host cell death induction (Fig. 7). In addition to PAMPs and DAMPs, bacterial circumvention of host inflammatory signaling becomes a cue that triggers host cell death to terminate bacterial infection. However, in the arms race between bacterial pathogens and host defense systems, the present data demonstrate that *Shigella* evolved strategies to overcome host cell death via effector-triggered immunity. Recent study has revealed that the complex interplay between NleE, NleB, and EspL effector of *Citrobacter rodentium* enables them to subvert innate immune signaling, apoptosis, necroptosis, and NLRP3 activation, suggesting that these infectious strategies seem to be a widely conserved among enteric bacterial pathogens (Yeap et al, 2025). Among approximately 50 *Shigella* effectors, only half have been functionally characterized at the molecular level, underscoring the importance of identifying and characterizing novel effectors to understand the interaction between effectors and host target molecules. In addition, the discovery of novel functions of previously identified bacterial effectors would provide information that may reveal new unidentified mechanisms underlying *Shigella* infection, which is critical to develop therapeutic approaches for the treatment of shigellosis.

# Methods

## Reagents and tools table

| Reagent/resource | Reference or source | Identifier or catalog number |
|---|---|---|
| **Experimental models** | | |
| *Shigella flexneri* YSH6000 | Sasakawa et al, 1986 | |
| *Shigella flexneri* S325 | Sasakawa et al, 1986 | |
| *Shigella flexneri ΔospC1* | Ashida et al, 2020 | |
| *Shigella flexneri ΔospC1/ospC1* | Ashida et al, 2020 | |
| *Shigella flexneri ΔospC1/ospC1-D172A* | This study | |
| *Shigella flexneri ΔospC1/ospC1-D226A* | This study | |
| *Shigella flexneri ΔospD3* | Ashida et al, 2020 | |
| *Shigella flexneri ΔospC1ΔospD3* | Ashida et al, 2020 | |
| *Shigella flexneri ΔospC1ΔospD3/ospC1-D172A* | This study | |
| *Shigella flexneri ΔospC1ΔospD3/ospC1-D226A* | This study | |
| *Shigella flexneri ΔospI* | Ashida et al, 2020 | |
| *Shigella flexneri ΔospC1ΔospI* | This study | |
| *Shigella flexneri ΔospC1ΔospI/ospI* | This study | |
| *Shigella flexneri ΔospC1ΔospI/ospI-C62A* | This study | |
| *Shigella flexneri ΔospD3ΔospI* | This study | |
| *Shigella flexneri ΔospD3ΔospI/ospI* | This study | |
| *Shigella flexneri ΔospD3ΔospI/ospI-C62A* | This study | |
| *Shigella flexneri ΔospC1ΔospD3ΔospI* | This study | |
| *Shigella flexneri ΔospC1ΔospD3ΔospI/ospC1* | This study | |
| *Shigella flexneri ΔospC1ΔospD3ΔospI/ospI* | This study | |
| *Shigella flexneri ΔospC1ΔospD3ΔospI/ospC1-ospI* | This study | |
| *Shigella flexneri ΔospG* | Ashida et al, 2020 | |
| *Shigella flexneri ΔospC1ΔospG* | This study | |
| HT-29 (*H. sapiens*) | ECACC | EC91072201-G0 |
| HeLa (*H. sapiens*) | ATCC | CRM-CCL-2 |
| HCT116 (*H. sapiens*) | ATCC | CCL-247 |
| T84 (*H. sapiens*) | ECACC | EC88021101-F0 |
| HT-55 (*H. sapiens*) | ECACC | EC85061105-F0 |
| 293 T (*H. sapiens*) | ATCC | CRL-3216 |
| Plat-A | Cell Biolabs Inc. | VPK-301 |
| HeLa/ospI | This study | |
| HeLa/ospI-CA | This study | |
| HT-29 cIAP1/2 KO | This study | |
| **Recombinant DNA** | | |
| pEGFP-h-*casp8* | This study | |
| pEGFP-m-*casp8* | This study | |
| pEGFP-h-*casp8 R430A* | This study | |
| pCMV-FLAG-*ospC1* | This study | |
| pCMV-FLAG-*ospC1-D172A* | This study | |
| pMX-puro-*ospI* | This study | |
| pMX-puro-*ospI-C62A* | This study | |
| **Antibodies** | | |
| Anti-RIP1 | Cell Signaling Technology | #3493 |
| Anti-p-RIP1 | Cell Signaling Technology | #44590 |
| Anti-RIP3 | Cell Signaling Technology | #15828 |
| Anti-p-RIP3 | Cell Signaling Technology | #93654 |
| Anti-TRADD | Cell Signaling Technology | #3694 |
| Anti-FADD | Cell Signaling Technology | #2782 |
| Anti-TNFR1 | Cell Signaling Technology | #3736 |
| Anti-Fas | Cell Signaling Technology | #4233 |
| Anti-c-IAP1 | Cell Signaling Technology | #7065 |
| Anti-c-IAP2 | Cell Signaling Technology | #3130 |
| Anti-Ubc13 | Cell Signaling Technology | #6999 |
| Anti-MLKL | Cell Signaling Technology | #14993 |
| Anti-p-MLKL | Cell Signaling Technology | #91689 |
| Anti-caspase-8 | Cell Signaling Technology | #4790 |
| Anti-caspase-8 | Cell Signaling Technology | #9746 |
| Anti-cleaved caspase-8 | Cell Signaling Technology | #9496 |
| Anti-caspase-3 | Cell Signaling Technology | #9662 |
| Anti-PARP | Cell Signaling Technology | #9532 |
| Anti-p-IκBα | Cell Signaling Technology | #2859 |
| Anti-IκBα | Cell Signaling Technology | #4814 |
| Anti-p-JNK | Cell Signaling Technology | #4668 |
| Anti-JNK | Cell Signaling Technology | #9252 |
| Anti-Poly/Mono-ADP Ribose | Cell Signaling Technology | #89190 |
| Anti-actin | MILLIPORE | #MAB1501 |
| Anti-GFP | MBL | 598 |
| TNF-α monoclonal antibody | Proteintech | 69002-1 |
| Anti-M2 FLAG | Sigma-Aldrich | F3165 |
| Anti-FLAG | Sigma-Aldrich | F7425 |
| Anti-GFP mAb-Agarose | MBL | D153-8 |

| Reagent/resource | Reference or source | Identifier or catalog number |
|---|---|---|
| Anti-Mouse IgG HRP | Sigma-Aldrich | A4416 |
| Anti-Rabbit IgG HRP | Sigma-Aldrich | A0545 |
| **Oligonucleotides and other sequence-based reagents** | | |
| Control siRNA 5′-CGUACGCGGAAUACUUCGAUU-3′ and 5′-UCGAAGUAUUCCGCGUACGUTT-3′ | SIGMA | |
| TNFR1 siRNA 5′-CCCUCAAAAUAAUUCGAUUtt-3′ and 5′-AAUCGAAUUAUUUUGAGGGtg-3′ | SIGMA | |
| Fas siRNA 5′-GGAAGACUGUUACUACAGUtt-3′ and 5′-ACUGUAGUAACAGUCUUCCtt-3′ | SIGMA | |
| Ubc13 #1 siRNA 5′-GCAUCAGUAUCUGACCUUUtt-3′ and 5′-AAAGGUCAGAUACUGAUGCtt-3′ | SIGMA | |
| Ubc13 #2 siRNA 5′-GGAAGAAUAUGUUUAGAUAtt-3′ and 5′-UAUCUAAACAUAUUCUUCCaa-3′ | SIGMA | |
| Ubc13 #3 siRNA 5′-CUUGAUUGUUGGAACCAAAtt-3′ and 5′-UUUGGUUCCAACAAUCAAGct-3′ | SIGMA | |
| TRADD siRNA 5′-GAUGCGCUGCGAAAUCGAUt-3′ and 5′-UCAGAUUUCGCAGCGCAUCct-3′ | SIGMA | |
| FADD siRNA 5′- CGGGAGUAGUUGGAAAGUUtt-3′ and 5′-AACUUUCCAACUACUCCCGca-3′ | SIGMA | |
| cIAP1 siRNA 5′-GGAUAACUGGAAACUAGGAtt-3′ and 5′-UCCUAGUUUCCAGUUAUCCag-3′ | SIGMA | |
| cIAP2 siRNA 5′-CACUCAUUACUUCCGGGUAtt-3′ and 5′-UACCCGGAAGUAAUGAGUGtg-3′ | SIGMA | |
| cIAP1 gRNA 5′-AUAUUCAACUUUCCCCGCCG-3′ | Invitrogen | |
| cIAP2 gRNA 5′-CGUAUUCCACUUUUCCUGCU-3′ | Invitrogen | |
| ospC1 cloning primer 5′-ccggaattcatgaatatatcagaaacactg-3′ and 5′-cgcggatccttaaatatatttattgtcaga-3′ | This study | |
| ospC1 D172A site-directed mutagenesis primer 5′-cgaaacacatacgctgatgctataaaaaaaatagctaatc-3′ and 5′-gattagctattttttttatagcatcagcgtatgtgtttcg-3′ | This study | |
| ospC1 D226A site-directed mutagenesis primer 5′-ggatatatgacattaaccgctcactttgataatgctattc-3′ and 5′-gaatagcattatcaaagtgagcggttaatgtcatatatcc-3′ | This study | |
| ospI primer 5′-ccggaattcatgattaatggggtgtcgtta-3′ and 5′-cgcggatcctcagcaaagcctcttactttt-3′ | This study | |
| ospI C62A site-directed mutagenesis primer 5′-ggtaacgcttcagggtgtgcgcttcat-3′ and 5′-ccctgaagcgttaccatccccaccaga-3′ | This study | |
| casp8 R430A site-directed mutagenesis primer 5′-GAATAACTGTGTTTCCTACGCAAACCCTGCAGAGGGAAC-3′ and 5′-GTTCCCTCTGCAGGGTTTGCGTAGGAAACACAGTTATTC-3′ | This study | |
| **Chemicals, enzymes, and other reagents** | | |
| Human TNF-α recombinant protein | PeproTech | #300-01A |
| Human IL-1β recombinant protein | PeproTech | #200-01B |
| phorbol 12-myristate 13-acetate | Sigma-Aldrich | P8139 |
| z-VAD-FMK | Calbiochem | 627610 |
| TNF-α inhibitor | Calbiochem | 654256 |
| Nec-1 | Sigma-Aldrich | N9037 |
| Apostatin-1 | Sigma-Aldrich | SML3028 |
| BV-6 | Selleckchem | S7597 |
| BAY11-7082 | Selleckchem | S2913 |
| TPCA-1 | Selleckchem | S2824 |

| Reagent/resource | Reference or source | Identifier or catalog number |
|---|---|---|
| ICCB-19 | Sigma-Aldrich | SML3006 |
| 5z-7 | MILLIPORE | 499610 |
| Cyclohexamide | Calbiochem | 239763 |
| McCoy's 5A | Gibco | 16600082 |
| Dulbecco's modified Eagle medium-high glucose | Sigma-Aldrich | D5796 |
| Dulbecco's modified Eagle's medium: Nutrient Mixture F-12 | Gibco | 11320033 |
| Eagle's minimum essential medium | Sigma-Aldrich | M4655 |
| MEM NEAA | Gibco | 11140050 |
| DO-TAP | Roche | 11202375001 |
| FuGENE 6 | Promega | E2691 |
| Lipofectamine RNAiMAX | Invitrogen | 13778150 |
| TrueCut™ Cas9 Protein | Invitrogen | A36497 |
| Lipofectamine™ CRISPRMAX™ Cas9 | Invitrogen | CMAX00001 |
| CytoTox96 Non-Radioactive Cytotoxicity Assay | Promega | G1780 |
| RealTime Glo Annexin V Apoptosis Assay | Promega | JA1000 |
| Cell Proliferation Kit I (MTT) | Roche | 11465007001 |
| Caspase-Glo-3/7 Assay Systems | Promega | G8091 |
| Caspase-Glo-8 Assay Systems | Promega | G8201 |
| human TNF-α ELISA kit | Invitrogen | 88-7346-88 |
| human IL-8 ELISA kit | Invitrogen | 88-8086-88 |
| ISOGEN II | Nippongene | 311-07361 |
| Gentamicin solution | Sigma-Aldrich | G1397 |
| Kanamycin sulfate solution | Wako | 117-00961 |
| **Software** | | |
| GraphPad Prism version 6 | Graphpad Software | https://www.graphpad.com |
| Illustrator | Adobe | https://www.adobe.com |
| Photoshop | Adobe | https://www.adobe.com |

## Strain and plasmids

*Shigella flexneri* strain YSH6000 was used as the WT, and S325 (*mxiA*::Tn5) was used as the T3SS-deficient negative control, as described previously (Sasakawa et al, 1986). Construction of non-polar mutants of *S. flexneri* YSH6000 was carried out using the red recombinase-mediated recombination system, as described previously (Datsenko and Wanner, 2000; Ashida et al, 2007). The *ospC1* and *ospI* coding sequences were amplified by PCR and cloned into pWKS130. Site-directed mutagenesis of *ospC1-D172A* (in which the aspartic acid residue at position 172 was substituted to alanine), *ospC1-D226A* (in which the aspartic acid residue at position 226 was substituted to alanine), and *ospI C62A* (in which the cysteine residue at position 62 was substituted to alanine), and *caspase-8 R430A* (in which the arginine residue at position 430 was substituted to alanine) was performed using the QuikChange site-directed mutagenesis kit (Stratagene).

## Materials

anti-RIP1 (#3493), anti-p-RIP1 (#44590), anti-RIP3 (#15828), anti-p-RIP3 (#93654), anti-TRADD (#3694), anti-FADD (#2782), anti-

TNFR1 (#3736), anti-Fas (#4233), anti-c-IAP1 (#7065), anti-c-IAP2 (#3130), anti-Ubc13 (#6999), anti-MLKL (#14993), anti-p-MLKL (#91689), anti-caspase-8 (#4790), anti-caspase-8 (#9746), anti-cleaved caspase-8 (#9496), anti-caspase-3 (#9662), anti-PARP (#9532), anti-p-IκBα (#2859), anti-IκBα (#4814), anti-p-JNK (#4668), anti-JNK (#9252), anti-Poly/Mono-ADP Ribose (#89190) (Cell Signaling Technology), anti-actin (#MAB1501, MILLIPORE), anti-GFP (598, MBL), anti-GFP mAb-Agarose (D153-8; MBL), TNF-α monoclonal antibody (69002-1; Proteintech), anti-M2 FLAG (F3165, Sigma-Aldrich) and anti-FLAG antibodies (F7425, Sigma-Aldrich) were obtained from the indicated suppliers. Human TNF-α recombinant protein (#300-01A; PeproTech), Human IL-1β recombinant protein (#200-01B; PeproTech), and phorbol 12-myristate 13-acetate (PMA; Sigma-Aldrich) were obtained from the indicated suppliers. The following inhibitors were obtained from indicated suppliers; z-VAD-FMK, and TNF-α inhibitor (#654256) (Calbiochem), Nec-1 and Apostatin-1 (Sigma-Aldrich), BV-6, BAY11-7082, TPCA-1 and ICCB-19 (Selleckchem), 5z-7 (MILLIPORE).

## Cell culture

Cells were maintained in 5% $CO_2$ at 37 °C. HT-29 and HCT116 cells were cultured in McCoy's 5A medium (Gibco) supplemented with 10% fetal calf serum. HeLa and 293T cells were cultured in Dulbecco's modified Eagle medium (Sigma) supplemented with 10% fetal calf serum. T84 cells were cultured in Dulbecco's modified Eagle's medium: Nutrient Mixture F-12 (Sigma) supplemented with 10% fetal calf serum. HT-55 cells were cultured in Eagle's minimum essential medium (Sigma) supplemented with 1% non-essential amino acids and 20% fetal calf serum. To construct HeLa cells stably expressing ospI or ospI-CA, cDNAs encoding these genes were subcloned into pMX-puro retroviral expression vectors. Retroviral supernatants were produced in Plat-A cells. HeLa cells were transduced with supernatants in the presence of DO-TAP (Roche), and then cloned under puromycin selection.

## Generation of knockout cell lines using the CRISPR/Cas9 system

cIAP1/2-double KO cells were generated using the TrueCut™ Cas9 Proteins System (Invitrogen) following the manufacturer's protocol. The Synthetic Guide RNAs against human cIAP1 or cIAP2 with the following sequences were prepared by Invitrogen: 5′-AUAUUCAACUUUCCCGCCG-3′ (cIAP1) and 5′-CGUAUUC-CACUUUUCCUGCU-3′ (cIAP2).

## Bacterial infection

Cells were infected with various *Shigella* strains expressing afimbrial adhesin (Afa) at a multiplicity of infection (MOI) of 10, as described previously (Ashida et al, 2010). Briefly, infection was initiated by centrifuging the plate at $400 \times g$ for 10 min. After incubation for 30 min at 37 °C, the cells were washed three times with PBS and transferred into fresh medium containing gentamicin (100 µg/mL) and kanamycin (60 µg/mL) to kill extracellular bacteria. The time of antibiotic treatment was defined as 0 h after infection. At the indicated times, infected cells were lysed and mixed with culture medium, precipitated with 10% trichloroacetic acid. All samples were separated by SDS-PAGE using either 7.5%, 10%, or 12% polyacrylamide gels, depending on the molecular weights of the target proteins, followed by immunoblotting.

## Cell death assay

Cells were infected with *Shigella* as described above, and then incubated at 37 °C. Aliquots of cellular supernatants obtained at the indicated time points were subjected to cytotoxicity assays. Cytotoxicity was analyzed using the CytoTox96 Non-Radioactive Cytotoxicity Assay (Promega). The following formula was used to calculate the amount of LDH release: [($OD_{490}$ sample release − $OD_{490}$-negative control release)/($OD_{490}$-positive control release − $OD_{490}$-negative control release)] × 100, where '$OD_{490}$ negative control release' represents the amount of LDH released into the culture supernatant from uninfected cells, and 'OD490 positive control release' represents the amount of LDH released after lysis of the uninfected cells. Apoptosis was assayed using the RealTime Glo Annexin V Apoptosis Assay (Promega). Cell viability (MTT assays) were performed with the Cell Proliferation Kit I (MTT) (Roche).

## Caspase activation assay

To measure caspase activity, cells were plated in each well of a 96-well plate and stimulated for 8 h with the indicated *Shigella* strains at an MOI of 10. Caspase-3/7 or -8 activities in cell lysates were analyzed using the Caspase-Glo-3/7 or -8 assays (Promega), respectively.

## Gentamicin protection assay

Cells were seeded into 24-well plates and infected with various *Shigella* strains. Briefly, infection was initiated by centrifuging the plate at $400 \times g$ for 10 min. After incubation for 30 min at 37 °C to allow bacterial uptake and invasion, the cells were washed thrice with PBS and transferred into fresh medium containing gentamicin (100 µg/mL) and kanamycin (60 µg/mL) to kill extracellular bacteria, and the cells were further incubated for 12 h. After incubation, the cells were washed thrice again with PBS and subsequently lysed with PBS containing 0.5% Triton X-100. The lysates were diluted with PBS and plated onto LB-agar plates. Finally, the number of intracellular bacteria was determined by calculating colony-forming units.

## RNAi

siRNAs against human TNFR1, Fas, Ubc13, TRADD, FADD, cIAP1 or cIAP2 with the following sequences, were prepared by Sigma: 5′-CCCUCAAAAUAAUUCGAUUtt-3′ and 5′-AAUCGAAUUAUUUU GAGGGtg-3′ (TNFR1), 5′-GGAAGACUGUUACUACAGUtt-3′ and 5′-ACUGUAGUAACAGUCUUCCtt-3′ (Fas), 5′-GCAUCAGUAUCU GACCUUUtt-3′ and 5′-AAAGGUCAGAUACUGAUGCtt-3′ (Ubc13 #1), 5′-GGAAGAAUAUGUUUAGAUAtt-3′ and 5′-UAUCUAAA CAUAUUCUUCCaa-3′ (Ubc13 #2), 5′-CUUGAUUGUUGGAAC CAAAtt-3′ and 5′-UUUGGUUCCAACAAUCAAGct-3′ (Ubc13 #3), 5′-GAUGCGCUGCGAAAUCUGAtt-3′ and 5′-UCAGAUUUCGCAG CGCAUCct-3′ (TRADD), 5′- CGGGAGUAGUUGGAAAGUUtt-3′ and 5′-AACUUUCCAACUACUCCCGca-3′ (FADD), 5′-GGAUAAC UGGAAACUAGGAtt-3′ and 5′-UCCUAGUUUCCAGUUAUCCag-3′

(cIAP1), 5′-CACUCAUUACUUCCGGGUAtt-3′ and 5′-UACCCGGA
AGUAAUGAGUGtg-3′ (cIAP2), and 5′-CGUACGCGGAAUACUUC
GATT-3′ and 5′-UCGAAGUAUUCCGCGUACGTT-3′ (luciferase as a
control). Cells were transfected using RNAiMax (Invitrogen). siRNA-
treated cells were used after 72 h for further analyses.

### RNA extraction and quantitative RT-PCR analysis

Total RNA was extracted using ISOGEN (Nippongene). First-strand
cDNA was synthesized from 0.5 μg total RNA with reverse transcriptase
using oligo(dT) primers. Real-time PCR was performed on cDNA
samples using a CFX96 (BIO-RAD) with SYBR Green system (TaKaRa).
The *GAPDH* expression levels were evaluated as an internal control. The
following primer pairs were used: human *IL8* (5′-CTGATTTCTG
CAGCTCTGTTG-3′ and 5′-GTCCACTCAATCACTCTCAG-3′), human
*GAPDH* (5′-TGCCCTCAACGACCACTTTG-3′ and 5′-TTCCTCTTG
TGCTCTTGCTGGG-3′).

### Immunoprecipitation

293T cells were transiently transfected using FuGENE 6 (Roche).
Cells were washed with PBS and lysed for 30 min at 4 °C in lysis
buffer containing 150 mM NaCl, 50 mM HEPES pH7.5, 1 mM
EDTA, 1% NP-40, and Complete protease inhibitor cocktail
(Roche). Lysates were cleared by centrifugation and proteins were
immunoprecipitated 2 h with anti-GFP agarose (MBL) at 4 °C.
Immunoprecipitates were washed five times with lysis buffer and
subjected to immunoblotting.

### Enzyme-linked immunosorbent assay (ELISA)

The cytokines released into the culture supernatants were
quantified using ELISA kits. The following ELISA kits were
purchased commercially: human TNF-α (Invitrogen, 88-7346-88)
and human IL-8 (Invitrogen, 88-8086-88). For IL-8 ELISA, ten-fold
diluted culture supernatant samples were used for measurement.

### Statistical analysis

Statistical analysis was performed in GraphPad Prism version 6.
Differences between two groups were evaluated using unpaired
two-tailed Student's *t* test. One-way ANOVA or two-way ANOVA
was used to analyze differences among multiple groups. *P*
values < 0.05 were considered significant.

## Data availability

This study includes no data deposited in external repositories.

The source data of this paper are collected in the following
database record: biostudies:S-SCDT-10_1038-S44318-025-00561-7.

Expanded view data, supplementary information, appendices are
available for this paper at https://doi.org/10.1038/s44318-025-00561-7.

## Peer review information

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

## Acknowledgements

This work was supported by a Grant-in-Aid for Scientific Research (B) (21H02728, 24K02276 to HA and 22H02865, 25K02492 to TS) and AMED under Grant Number 20gm6010009h0004 (HA). Part of this work was supported by grants from the Naito Foundation (HA), a research grant from the Astellas Foundation for Research on Metabolic Disorders (HA), the Uehara Memorial Foundation (HA), GSK Japan Research Grant 2016 (HA), the Kawano Masanori Memorial Foundation for Promotion of Pediatrics (HA), TERUMO FOUNDATION for LIFE SCIENCES and ARTS (HA), the Senri Life Science Foundation (HA), the Hamaguchi Foundation for the Advancement of Biochemistry (HA), the Japan Foundation for Pediatric Research (HA), Mishima Kaiun Memorial Foundation (HA), OHYAMA HEALTH FOUNDATION Inc. (HA), Asahi Glass Foundation (HA), G-7 Scholarship Foundation (HA), the Chemo-Sero-Therapeutic Research Institute (HA), SHIONOGI INFECTIOUS DISEASE RESEARCH PROMOTION FOUNDATION (HA), Nagase Science and Technology Foundation (HA), the Waksman foundation of Japan Inc. (HA) and the Takeda Science Foundation (HA).

## Author contributions

**Hiroshi Ashida**: Conceptualization; Resources; Data curation; Formal analysis; Supervision; Funding acquisition; Validation; Investigation; Visualization; Methodology; Writing—original draft; Project administration; Writing—review and editing. **Tokuju Okano**: Resources; Investigation. **Tamako Iida**: Resources; Investigation. **Poramed Onsoi**: Resources; Investigation. **Chihiro Sasakawa**: Resources. **Toshihiko Suzuki**: Supervision; Writing—review and editing.

Source data underlying figure panels in this paper may have individual authorship assigned. Where available, figure panel/source data authorship is listed in the following database record: biostudies:S-SCDT-10_1038-S44318-025-00561-7.

## Disclosure and competing interests statement

The authors declare no competing interests.

# Expanded View Figures

**Figure EV1.  The *Shigella* effector OspI triggers caspase-8 activation.**

(**A, B**) HeLa cells stably expressing OspI or OspI-C62A were stimulated with TNF-α (25 ng/mL), IL-1β (50 ng/mL), or PMA (50 μg/mL) and incubated for 8 h. Caspase-8 activity (**A**) or caspase-3 activity (**B**) was measured and is reported as relative light units (RLU) of stimulated samples normalized to the value in unstimulated samples. (**A**) Data are expressed as the mean ± SD from quadruplicate and representative of three independent experiments ($P$ values: $P < 0.0001$ (left) and $P < 0.0001$ (right); two-way ANOVA). (**B**) Data are expressed as the mean ± SD from triplicate and representative of three independent experiments ($P$ values: $P < 0.0001$ (left) and $P < 0.0001$ (right); two-way ANOVA). (**C**) HeLa cells stably expressing OspI or OspI-C62A were stimulated with TNF-α (25 ng/mL). At the indicated time points, cells were harvested and cell lysates were subjected to immunoblotting. Data are representative of three independent experiments. (**D**) HeLa cells stably expressing OspI or OspI-C62A were stimulated with TNF-α (10 ng/mL) or PMA (50 μg/mL) and incubated for 1 h. The expression level of the *IL8* gene was determined by real-time PCR. Relative expression change in gene expression was calculated using unstimulated cells as the control. Data are expressed as the mean ± SD from three independent experiments ($P = 0.8172$ for TNF-α and $P < 0.0001$ for PMA; two-way ANOVA). Data are considered significant when $P < 0.05$, with *$P < 0.05$ or n.s., not significant (**A, B, D**).

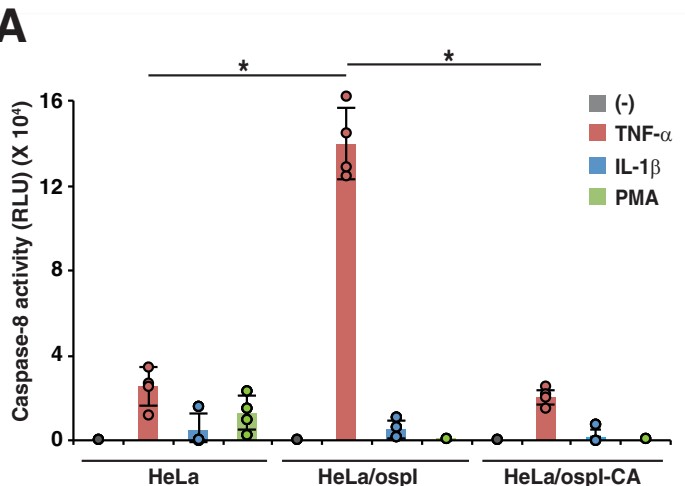

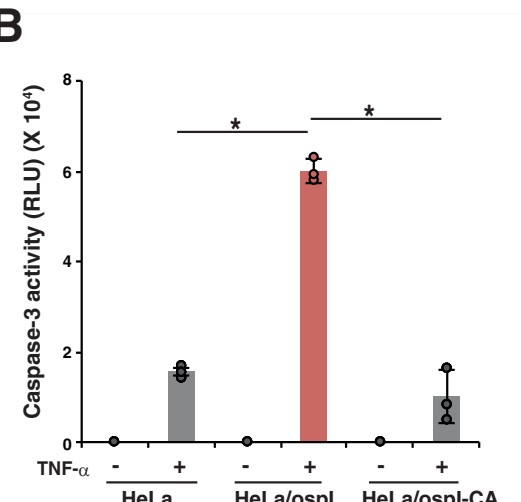

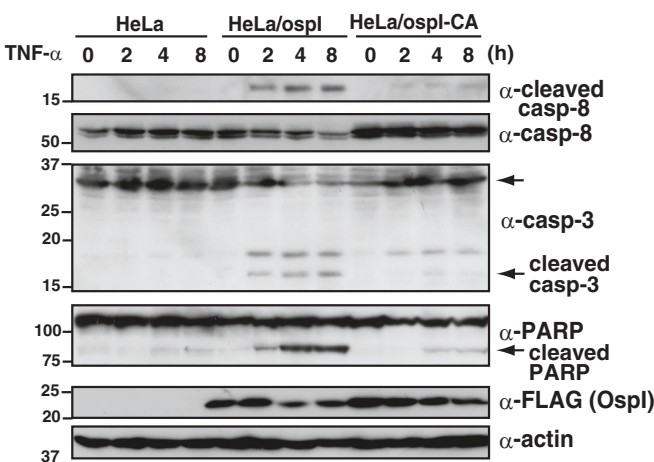

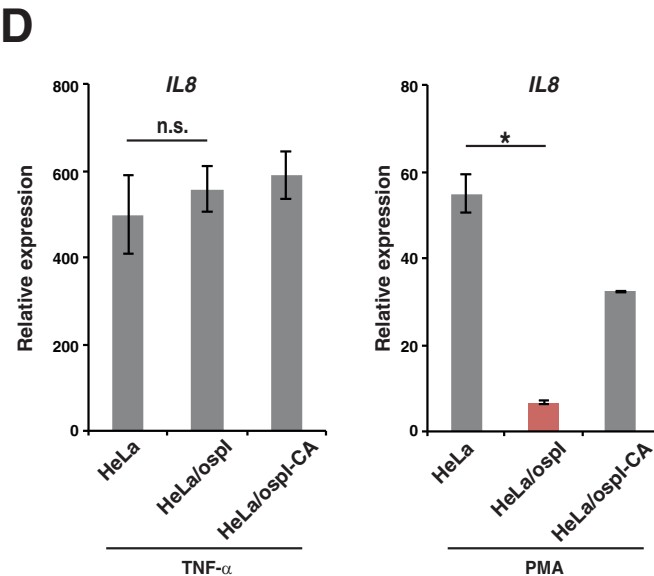

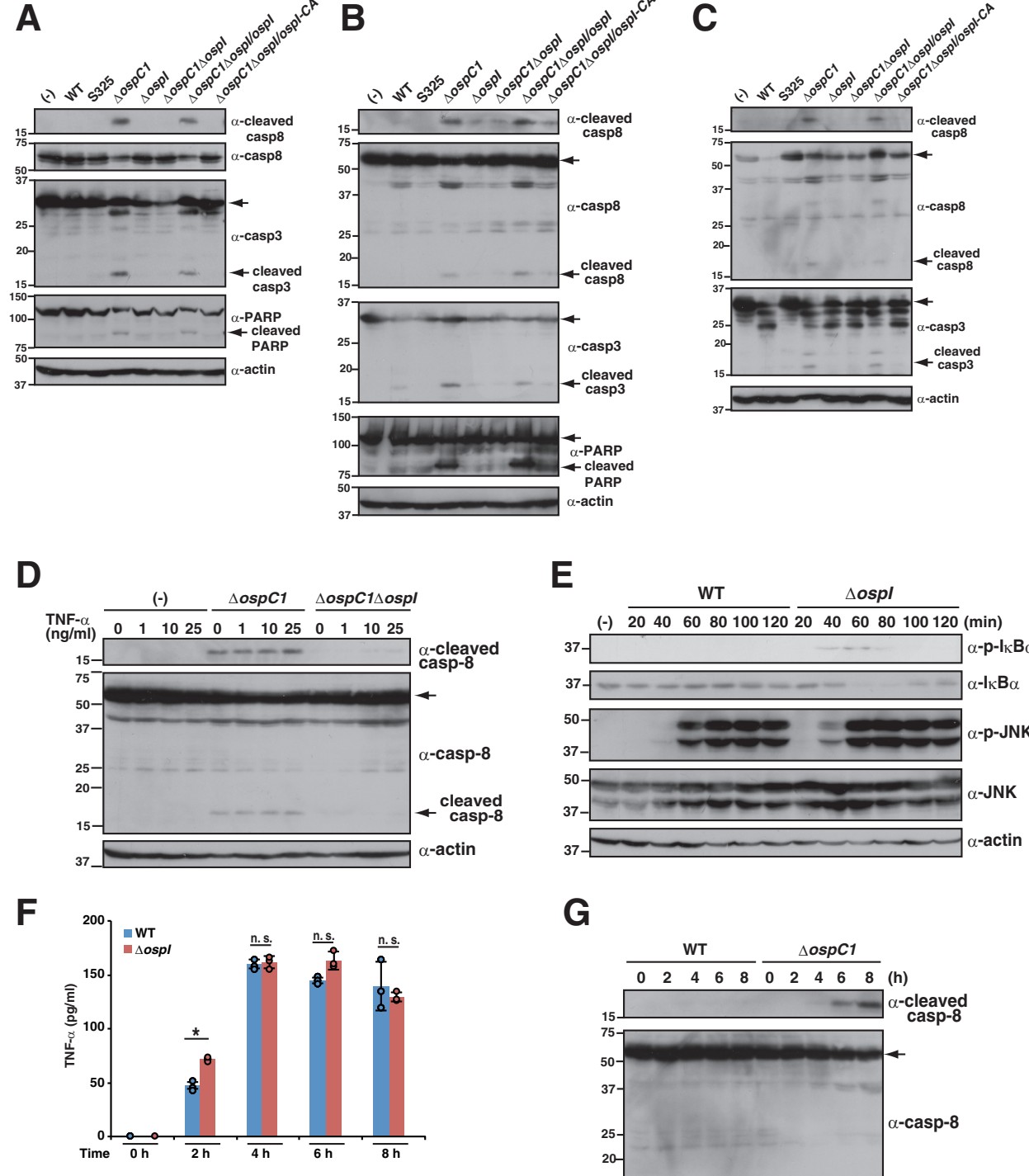

**Figure EV2. The *Shigella* effector OspI triggers caspase-8 activation.**

(A–C) HeLa (A), HCT116 (B), or T84 (C) cells were infected with the indicated *Shigella* strains and incubated for 8 h (A), 6 h (B), or 12 h (C). Cell lysates were subjected to immunoblotting. (D) HT-29 cells were infected with the indicated *Shigella* strains in the presence or absence of TNF-α and incubated for 8 h. Cell lysates were then subjected to immunoblotting. (E–G) HT-29 cells were infected with the indicated *Shigella* strains. Cell lysates (E, G) or culture supernatants (F) obtained at the indicated time points were subjected to immunoblotting (E, G) or ELISA (F). (F) Data are expressed as the mean ± SD from triplicate and representative of three independent experiments (TNF-α : $P = 0.0384$ for 2 h, $P > 0.9999$ for 4 h, $P = 0.2229$ for 6 h, and $P = 0.8763$ for 8 h. IL-8: $P = 0.4407$ for 4 h, $P = 0.0129$ for 6 h, and $P < 0.0001$ for 8 h; two-way ANOVA). Data are representative of three independent experiments (A–E, G). Molecular weights in immunoblots are in kDa. Data are considered significant when $P < 0.05$, with *$P < 0.05$ or n.s., not significant (F).

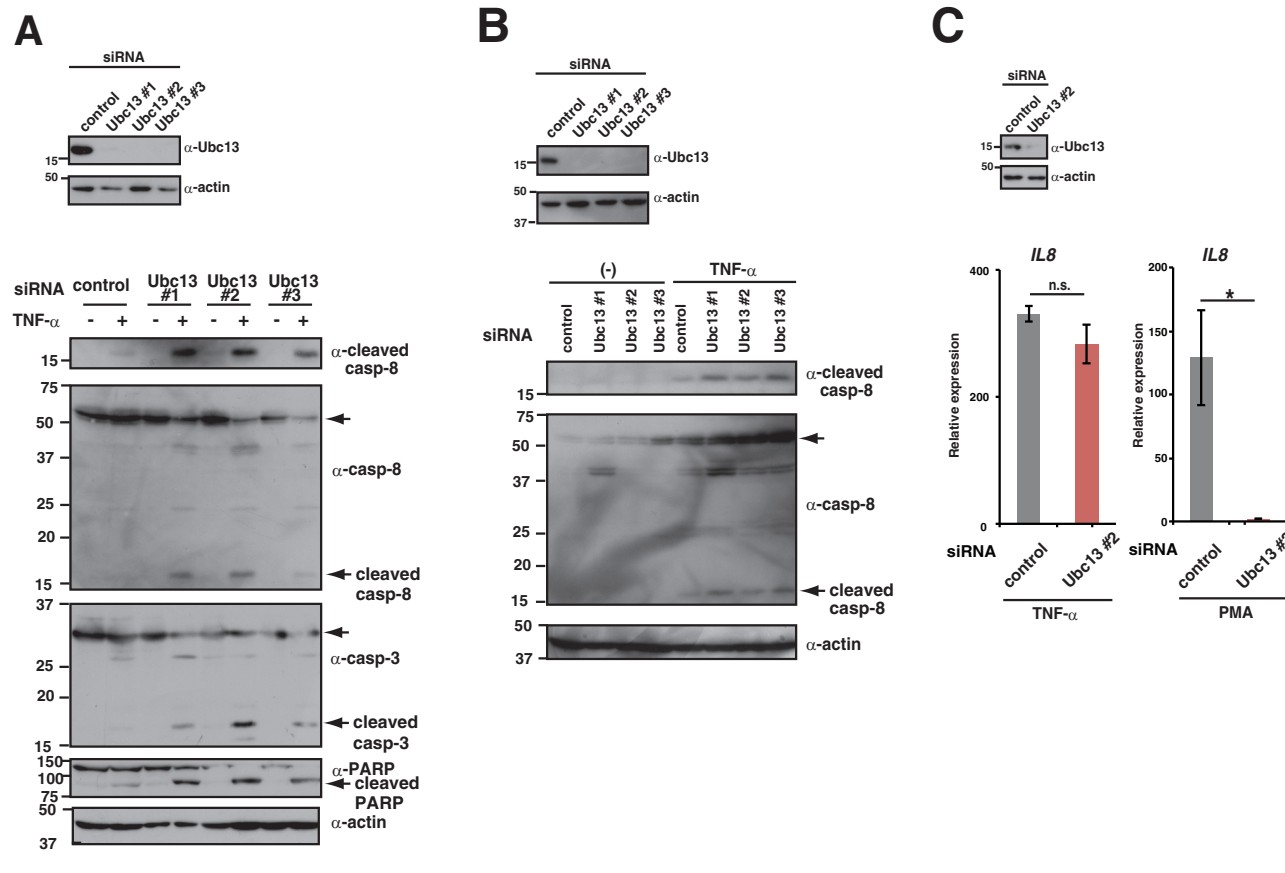

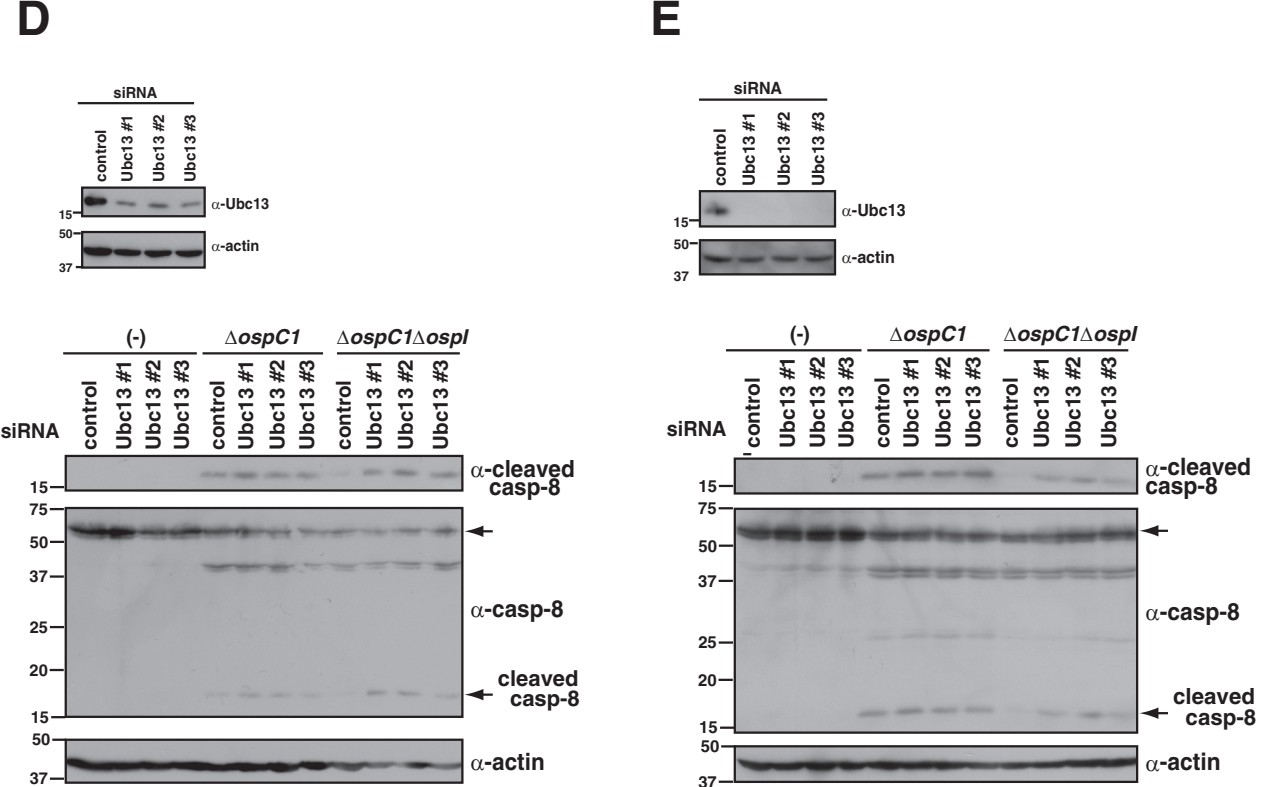

**Figure EV3. Ubc13 inactivation triggers caspase-8 activation under TNF-α stimulation.**

(A, B) HeLa (A) or HCT116 (B) cells treated with control or Ubc13 siRNAs were stimulated with TNF-α (25 ng/mL) and incubated for 8 h. Cell lysates were subsequently subjected to immunoblotting. The knockdown efficiency of the indicated siRNAs was assessed by immunoblotting (inset). (C) HeLa cells treated with control or Ubc13 siRNAs were stimulated with TNF-α (10 ng/mL) or PMA (50 μg/mL) and incubated for 1 h. The expression level of the *IL8* gene was determined by real-time PCR. Relative expression change in gene expression was calculated using unstimulated cells as the control. Data are expressed as the mean ± SD from three independent experiments ($P = 0.2176$ for TNF-α and $P = 0.0278$ for PMA; two-tailed Student's *t* test). The knockdown efficiency of the indicated siRNAs was assessed by immunoblotting (inset). (D, E) HeLa (D) or HCT116 (E) cells treated with control or Ubc13 siRNAs were infected with the indicated *Shigella* strains and incubated for 8 h. Cell lysates were subjected to immunoblotting. The knockdown efficiency of the indicated siRNAs was assessed by immunoblotting (inset). Data are representative of three independent experiments (A, B, D, E). Molecular weights in immunoblots are in kDa. Data are considered significant when $P < 0.05$, with *$P < 0.05$ or n.s., not significant (C).

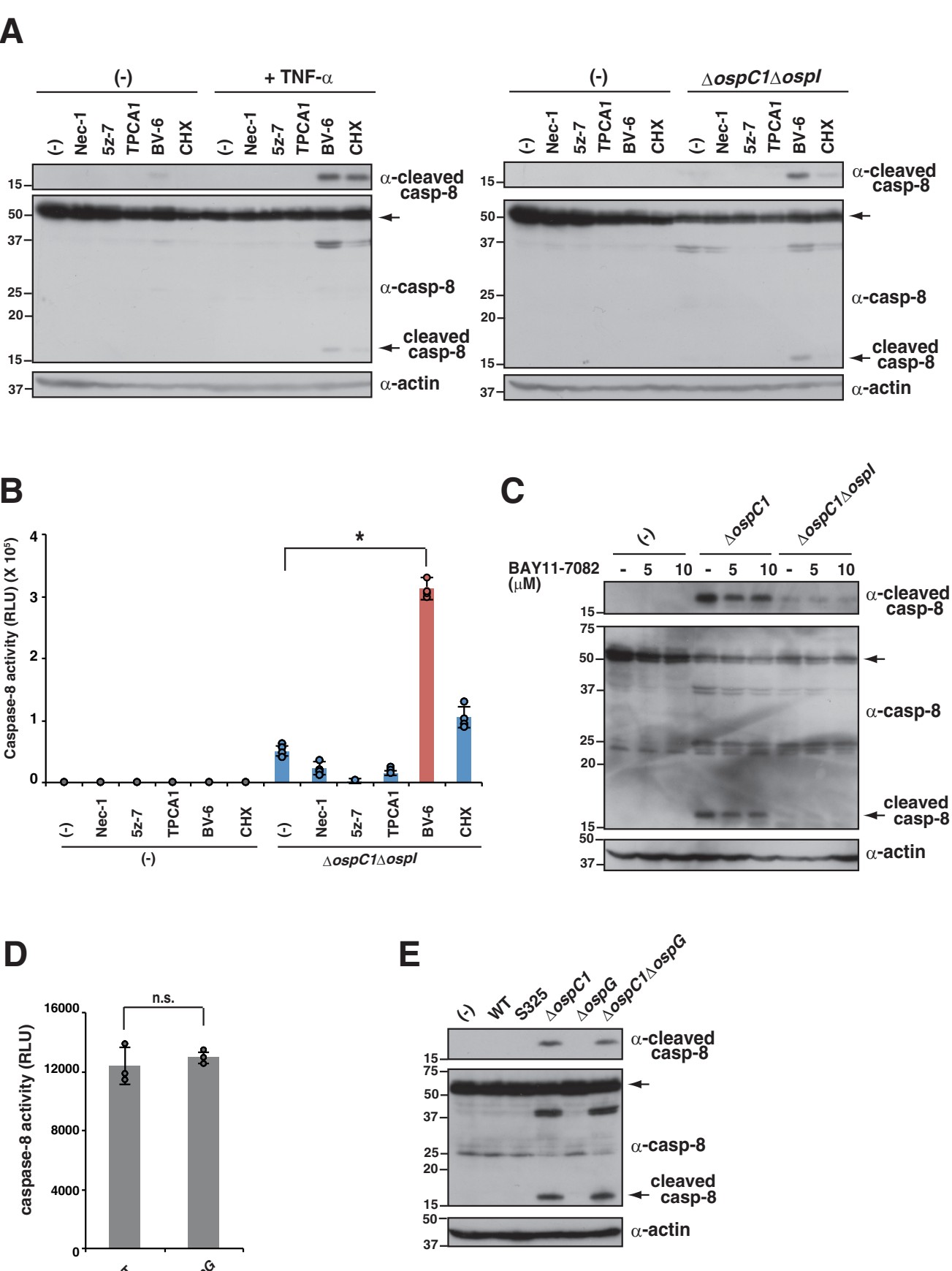

**Figure EV4. *Shigella* effector OspI-mediated IAP inactivation triggers caspase-8 activation.**

(A, B) HT-29 cells were stimulated with TNF-α (25 ng/mL) or infected with *Shigella ΔospC1ΔospI* in the presence or absence of Nec-1 (50 μM), 5z-7 (0.5 μM), TPCA1 (5 μM), BV-6 (1 μM), or CHX (25 μg/mL) and incubated for 8 h. Cell lysates were subjected to immunoblotting (**A**) or measurement of caspase-8 activity (**B**). Caspase-8 activity is reported in terms of relative light units (RLUs) of infected samples normalized to the values in uninfected samples. Data are expressed as the mean ± SD from quadruplicate and representative of three independent experiments (*P* value: *P* < 0.0001; two-way ANOVA). (**C**) HT-29 cells were infected with the indicated *Shigella* strains in the presence or absence of a BAY11-7082 inhibitor (5 or 10 μM) and incubated for 8 h. Cell lysates were subjected to immunoblotting. (**D**) HT-29 cells were infected with the indicated *Shigella* strains and incubated for 8 h. Cells were then harvested and caspase-8 activity was measured. Caspase-8 activity is reported as RLUs of infected samples normalized to the values in uninfected samples. Data are expressed as the mean ± SD from triplicate and representative of three independent experiments (*P* = 0.52; two-tailed Student's *t* test). (**E**) HT-29 cells were infected with indicated *Shigella* strains and incubated for 8 h. Cell lysates were subjected to immunoblotting. Data are representative of three independent experiments (**A, C, E**). Molecular weights in immunoblots are in kDa. Data are considered significant when *P* < 0.05, with *\*P* < 0.05 or n.s., not significant (**B, D**).

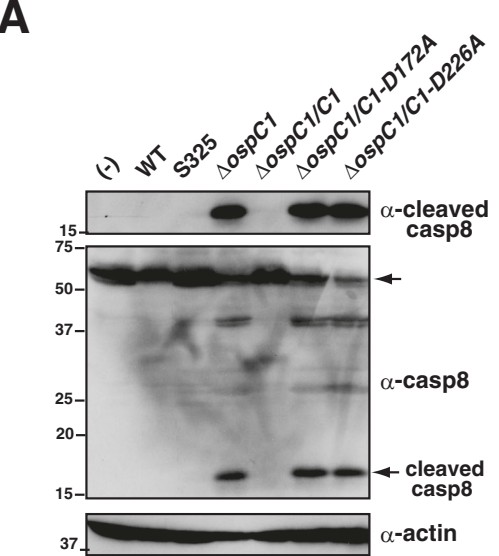

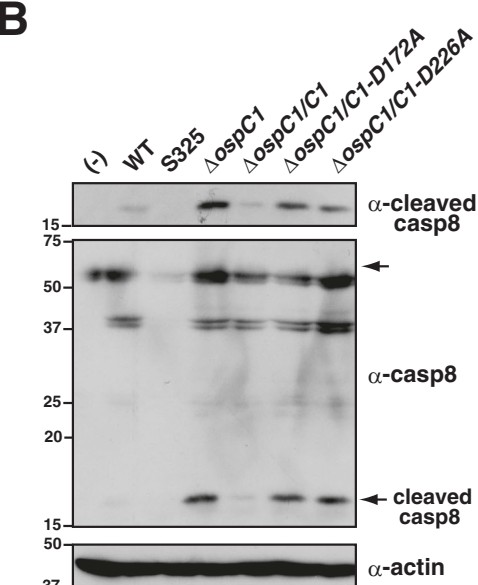

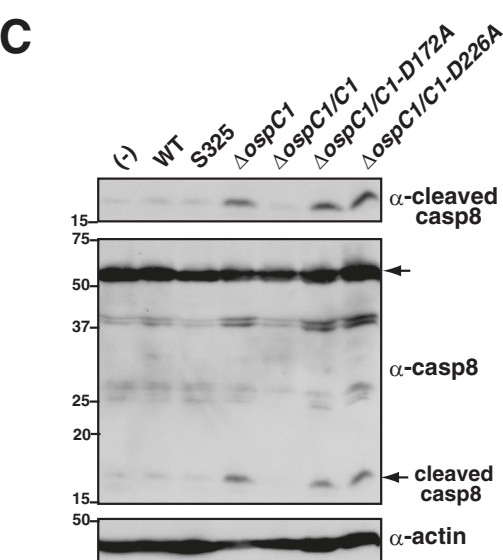

**Figure EV5.  The ADP-riboxanation activity of OspC1 is required for caspase-8 inhibition.**

(A–C) HeLa (A), HCT116 (B) or T84 (C) cells were infected with *Shigella* WT, S325, *ΔospC1*, *ΔospC1/ospC1* (*ΔospC1* complemented with *ospC1*), *ΔospC1/ospC1-D172A* (*ΔospC1* complemented with *ospC1-D172A*), or *ΔospC1/ospC1-D226A* (*ΔospC1* complemented with *ospC1-D226A*) strains and incubated for 6 h (A, B) or 12 h (C). Cell lysates were subjected to immunoblotting. All data are representative of three independent experiments. Molecular weights in immunoblots are in kDa.

# A

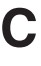

| TNF-α | - | + | + | + | + | + |
|---|---|---|---|---|---|---|
| BV6 | - | - | + | + | - | - |
| CHX | - | - | - | - | + | + |
| z-VAD | - | - | - | + | - | + |

α-p-MLKL (50)

α-MLKL (50)

α-cleaved casp8 (15)

α-casp8 (←, 50, 37, 25, 20)

← cleaved casp-8 (15)

α-casp8 (50)

α-p-RIPK1 (75)

α-RIPK1 (75)

α-p-RIPK3

α-RIPK3 (50)

α-actin (37)

# B

Cytotoxicity (%) plotted against: (-), TNF-α, TNF-α+BV-6, TNF-α+BV-6+z-VAD, TNF-α+CHX, TNF-α+CHX+z-VAD. Significance markers * between TNF-α+BV-6 and TNF-α+BV-6+z-VAD, and between TNF-α+CHX and TNF-α+CHX+z-VAD.

# C

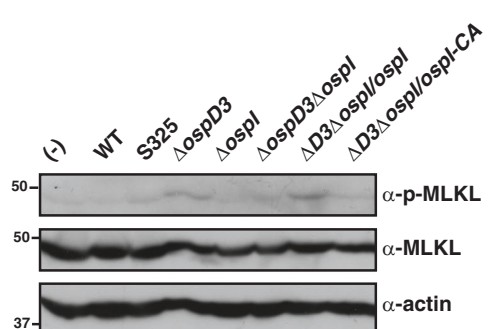

Lanes: (-), WT, S325, ΔospD3, ΔospI, ΔospD3ΔospI, ΔD3ΔospI/ospI, ΔD3ΔospI/ospI-CA

α-p-MLKL (50)

α-MLKL (50)

α-actin (37)

**Figure EV6. Caspase-8 is the molecular switch for apoptosis and necroptosis.**

(A, B) HT-29 cells were treated with TNF-α (25 ng/mL) plus the indicated inhibitors (BV-6, 1 µM; CHX, 25 µg/mL; Z-VAD-fmk, 10 µM) and incubated for 8 h. Cell lysates and aliquots of cellular supernatants were subjected to immunoblotting (A) or cytotoxicity assays (B), respectively. (B) Data are expressed as the mean ± SD from triplicate and representative of three independent experiments (*P* values: *P* < 0.0001 (left) and *P* < 0.0001 (right); two-way ANOVA). (C) HT-55 cells were infected with the indicated *Shigella* strains and incubated for 12 h before cell lysates were subjected to immunoblotting. Data are representative of three independent experiments (A, C). Molecular weights in immunoblots are in kDa. Data are considered significant when *P* < 0.05, with **P* < 0.05 (B).

