## [Peer Review File · The EMBO Journal]

***Shigella* type-III secretion system effectors counteract the induction of host inflammation and cell death**

Hiroshi Ashida, Tokuju Okano, Tamako Iida, Poramed Onsoi, Chihiro Sasakawa, and Toshihiko Suzuki

Corresponding author(s): Hiroshi Ashida (ashi.bact@tmd.ac.jp) , Toshihiko Suzuki (suzuki.bact@tmd.ac.jp)

Review Timeline:

Submission Date:	10th Nov 24
Editorial Decision:	18th Dec 24
Revision Received:	13th Jun 25
Editorial Decision:	5th Aug 25
Revision Received:	10th Aug 25
Accepted:	14th Aug 25

Editor: Ioannis Papaioannou

Transaction Report:

Dear Dr. Ashida,

Thank you again for submitting your manuscript EMBOJ-2024-119594 for consideration by The EMBO Journal, and for your patience during peer review. Your manuscript has now been seen by four experts in the field, and we have received the full set of their well-informed and detailed reports, which are included below.

As you will see, the feedback we received from the referees on your manuscript is mixed. All referees recognize that this is an interesting and relevant topic, but they -especially refs. #1 and #2- also identify major limitations in the study and raise concerns regarding the in vivo relevance of the results (given that all experiments were performed in cell lines with unclear relevance), the robustness of some conclusions (that were based on the use of inhibitors and single siRNA knockdown experiments rather than more rigorous genetic approaches), and the conceptual novelty of the results. The referees also provide lists of detailed suggestions for strengthening the work and the manuscript further.

Given these major concerns, I regret to say that your manuscript as it stands is not sufficiently developed for publication in The EMBO Journal. However, and in light of the potential interest and significance of your findings, we would be open to reconsidering a thoroughly revised and substantially strengthened version of your manuscript should you be willing and able to address the major concerns raised by the referees by expanding your study to more native relevant systems and providing further experimental support to the conclusions of your study. I should add that the outcome of this process cannot be guaranteed, as it depends on the completeness of your responses in the revised version, and the extent to which the referees will find the provided advance sufficient for publication in The EMBO Journal.

If you decide to embark on such a thorough revision for resubmission to The EMBO Journal, please submit along with your revised manuscript a detailed point-by-point response addressing all referees' comments. We generally allow three months as standard revision time (March 17, 2025), but an extension to up to 6 months might be possible should you need more time to sufficiently address the referees' concerns. Please let us know if you foresee a problem in meeting the initial three-month deadline.

As a matter of policy, competing manuscripts published during this period will not negatively impact our assessment of the conceptual advance presented by your study. However, we request that you contact us as soon as possible upon publication of any related work, to discuss how to proceed.

Thank you for the opportunity to consider your work for publication in The EMBO Journal. I look forward to your revision.

Best regards,

Ioannis

Instructions for preparing your revised manuscript

1. When you are ready to submit the revision, please upload:

- A Word file of the manuscript text (including legends of main Figures, EV Figures and Tables). Please make sure that changes are highlighted (or "tracked") to be clearly visible.

- Individual production-quality figure files (one file per figure). When assembling your figures, please refer to our figure preparation guidelines in order to ensure proper formatting and readability in print as well as on screen:

If the data shown in a figure are obtained from n {less than or equal to} 2, please use scatter plots showing the individual data points.

i. the name of the statistical test used to generate error bars and P values

ii. the number (n) of independent experiments (please specify technical or biological replicates) underlying each data point

(discussion of statistical methodology can be reported in the Materials and Methods section, but figure legends should contain a basic description of n, P, and the test applied)

iii. the nature of the bars and error bars (s.d., s.e.m.).

- A point-by-point response to the referees' comments, with a detailed description of the changes made (as a word file). All referees' concerns must be fully addressed and their suggestions taken on board. When preparing your letter of response to the referees' comments, please bear in mind that this will form part of the Review Process File and will therefore be available online to the community. Please note that you have the possibility to opt out of the transparent process at any stage prior to publication by letting the editorial office know (contact@embojournal.org); if you do opt out, the Review Process File link will point to the following statement: "No Review Process File is available with this article, as the authors have chosen not to make the review process public in this case.". For more details on our Transparent Editorial Process, please visit our website: <https://www.embopress.org/page/journal/14602075/authorguide#transparentprocess>

- Expanded View (EV) files (replacing Supplementary Information) that are collapsible/expandable online. A maximum of 5 EV Figures can be typeset. EV Figures should be cited as "Figure EV1, Figure EV2" etc. in the text, and their respective legends should be included in the manuscript file after the legends of regular figures. See detailed instructions regarding Expanded View files here:

- For the figures that you do NOT wish to display as Expanded View figures, they should be bundled together with their legends in a single PDF file called "Appendix", which should start with a short Table of Contents (including page numbers). Appendix figures should be referred to in the main text as: "Appendix Figure S1, Appendix Figure S2" etc. Please see detailed instructions here: <https://www.embopress.org/page/journal/14602075/authorguide#expandedview>

- A complete author checklist, which you can download from our author guidelines (<https://www.embopress.org/page/journal/14602075/authorguide>). Please note that the checklist will also be part of the Review Process File.

2. Please note that no statistics should be calculated and shown in Figures if $n=2$. Please also note that each p value should be reported as an exact value.

3. Before submitting your revision, primary datasets (and computer code, where appropriate) produced in this study need to be deposited in appropriate public databases (see <https://www.embopress.org/page/journal/14602075/authorguide#dataavailability>).

The accession numbers, databases, and the specific URLs (links) should be listed in a formal "Data availability" section (placed after Methods), following the example below:

"The RNA-seq datasets produced in this study are available in the following database:

Gene Expression Omnibus GSE46843 (<https://www.ncbi.nlm.nih.gov/geo/query/acc.cgi?acc=GSE46843>)"

*** All links should resolve to a page where the data can be accessed. ***

*** Please remember to provide in the Data availability section of your revised manuscript reviewer passwords if the datasets are not yet public. ***

*** The Data Availability Section is restricted to new primary data that are part of this study. In case you have no data that require deposition in a public database, please state so instead of referring to the database: "Our study includes no data deposited in public repositories." under the heading "Data availability". ***

4. Please check that the title and the abstract of the manuscript are brief, yet explicit, even to non-specialists. The length of the title should not exceed 100 characters, and the abstract should be a single paragraph not exceeding 175 words.

5. Please also note our reference format: <https://www.embopress.org/page/journal/14602075/authorguide#referencesformat>.

7. Please remember: digital image enhancement is acceptable practice, as long as it accurately represents the original data and conforms to community standards. If a figure has been subjected to significant electronic manipulation, this must be noted in the figure legend or in the "Materials and Methods" section. The editors reserve the right to request original versions of figures and the original images that were used to assemble the figure.

8. Our journal encourages inclusion of data citations in the reference list to directly cite datasets that were obtained from public databases. Data citations in the article text are distinct from normal bibliographical citations and should directly link to the database records from which the data can be accessed. In the main text, data citations are formatted as follows: "Data ref: Smith et al, 2001" or "Data ref: NCBI Sequence Read Archive PRJNA342805, 2017". In the Reference list, data citations must be labeled with "[DATASET]". A data reference must provide the database name, accession number/identifiers, and a resolvable link to the landing page from which the data can be accessed at the end of the reference. Further instructions are available at: <https://www.embopress.org/page/journal/14602075/authorguide#referencesformat>.

9. We request authors to consider both actual and perceived competing interests. Please review our policy (<https://www.embopress.org/page/journal/14602075/authorguide#conflictsinterest>) and update your competing interests statement if necessary. Please name this section 'Disclosure and competing interests statement' and place it after the Acknowledgements section.

10. Please note that all corresponding authors are required to provide an ORCID ID upon submission of a revised manuscript (<https://orcid.org/>). Please find instructions on how to link your ORCID ID to your account in our manuscript tracking system in our Author guidelines (<https://www.embopress.org/page/journal/14602075/authorguide#authorshipguidelines>).

11. We use CRediT to specify the contributions of each author in the journal submission system. CRediT replaces the author contribution section, which should be removed from the manuscript. Please use the free text box to provide more detailed descriptions. See also guide to authors: <https://www.embopress.org/page/journal/14602075/authorguide#authorshipguidelines>.

13. We would also welcome the submission of cover suggestions or motifs to be used by our Graphics Illustrator in designing a cover.

14. Please use the link below to submit your revision:
<https://emboj.msubmit.net/cgi-bin/main.plex>

Referee #1:

In their manuscript, the authors have characterised the infection of Shigella strains in HT29 and HeLa cells in vitro. This is a severe limitation as it is not clear whether this also happens in primary tissues during native infection. Using a series of well-controlled experiments, the authors have investigated single effector mutants that block different steps of inter-connected programmed cell pathways. These include RIPK1-caspase-8-dependent apoptosis and RIPK1-RIPK3-dependent necroptosis. They use single mutants of ospC1 (a caspase-1,3,4,8 inhibitor), ospI (inhibitor of Ube13, a ubiquitin conjugating enzyme) and ospD3 (a protease that inactivates RIPK1 and RIPK3). Every effector they have studied here has been extensively characterised before, and the main novelty is looking at them together, but even that has been done previously by the same group (0.15252/embj.2020104469).

Their experiments suggest that Shigella infection leads to interference at distinct steps that affect programmed cell death pathways. While it is nice to see the three effectors compared side-by-side, much of this is expected from previous work, and indeed discussed elsewhere based on the known blockades that trigger alternative signalling (e.g., caspase-8 blockade is required for necroptosis, NF- κ B blockade does influence apoptosis, among other effects). I also do not agree this is effector-triggered immunity; this merely shows that these effectors work together as part of a network, which is also the case with other Gram-negative enterics (e.g., Salmonella, EPEC/Citrobacter). Therefore, overall, the advance is limited. As far as I can tell, the main new findings are the specific effect of Ube13 on IAPs and OspC1 on caspase-8 modification and activity. It is also likely that their findings only apply to these specific cancer cell lines, which prevents generalisation of these findings to the actual human infection. In particular, HT29 is a Caspase-8 heterozygous line (one wildtype and a p.Q406* copy, which affects caspase-8 functions), and therefore has unique programme cell death features not seen in 'wildtype' homozygous cells (10.1158/0008-5472.CAN-14-0013; 10.1158/0008-5472.CAN-12-3342). HeLa cells (which are not intestinal in origin) do not express RIPK3 and therefore naturally do not undergo necroptosis; therefore, they are not suitable for studies on the interconnectedness between apoptosis and necroptosis. It is also not clear why some experiments are performed in HT29 and some in HeLa and no scientific rationale for this is provided (e.g., Why are several OspI experiments in HeLa?). Given that most cancer cells have unusual cell death signalling, much of this work needs to be carried out in primary intestinal cells (e.g., organoids) or more native systems to be convincing and generalisable to natural infection by Shigella. Other inconsistencies and lack of support for some of the conclusions detailed below:

Major points.

1. Unless these findings are shown in more native systems, much of the work is not novel and a repeat of what has previously been shown with individual effectors or families of effectors. Much of the work in HT29 should be repeated at least in an intestinal epithelial cell line (there are several - refer to the CRC database <http://www.coloncanceratlas.org/>) that has no mutations (at least) in the CASP8 gene and others in the pathways being addressed.
2. To claim TNFR1 is driving death, did these cells produce TNF during infection - either in the supernatant or membrane-bound? Have the authors tested anti-TNF antibodies instead of a small molecule inhibitor, which can have off-target effects?
3. There are inconsistencies as well - why is TNF-dependent IL18 production not affected by *ospl* in Fig EV1? This is not in line with findings elsewhere in the paper.
4. To convincingly show TRADD/FADD-independent apoptosis, it would be good to also show whether the silencing did reduce TNF or Fas-dependent apoptosis in these settings. Without this it is plausible that low levels of these adaptors despite RNAi suffice for cell death.
5. The authors test various mutants of *Shigella*, but do not demonstrate that the level of infection is comparable across strains. This is an important control experiment.
6. It would be informative to compare relative activity of Cas8 during infection conditions and TNF treatments +/- RIPK1 inhibitor.

Minor points

1. The authors use 'loose' language in several places (e.g., line 193, 204), such as "*ospl* induces caspase-8...", this is not accurate as it implies a direct action. We know it is very indirect and the authors should only state an "*ospl* mutant leads to ..." or "loss of *ospl* results in..."
2. On line 171 - do not say "prove this hypothesis", hypothesis should only be tested and results reported.

Referee #2:

This interesting paper investigates how *Shigella* is sensed by the innate immune system. *Shigella* is a very important bacterial pathogen that remains poorly understood. The question of how *Shigella* is sensed by the innate immune system is an important question to address. The authors propose *Shigella* is sensed via an "effector-triggered" immune response. This response is induced by a secreted *Shigella* effector called *OspI* that inhibits NF- κ B signaling. Inhibition of NF- κ B signaling presumably evolved to benefit the bacteria, but in response, the host has evolved to sense that its NF- κ B pathway has been disrupted and respond by activating a Caspase-8-dependent cell death response. Interestingly, another *Shigella* effector, *OspC1*, has evolved to inhibit Caspase-8. Thus there is a fascinating attack and counterattack (or "arms race") between *Shigella* and its human hosts.

Many of the key aspects of this paper were previously known. For example, it was known that (a) *OspI* inhibits NF- κ B, (b) inhibition of NF- κ B signaling alters TNF signaling from a pro-survival to a pro-death output, (c) *OspC1* was already shown to block Caspase-8. The main novel contributions of the manuscript are to show that Caspase-8 activation induced by *Shigella* depends on inhibition of NF- κ B by *OspI*, and that the *OspI*-induced Caspase-8 activation also depends on signaling via the TNF receptor. These observations allow the authors to piece together previously established findings in the field into an interesting overall model that is an important contribution to our understanding of the host response to *Shigella*.

The main limitation of the paper is that the experimental methods are not state-of-the-art. There is a lot of reliance on inhibitors and siRNAs (often just a single siRNA) instead of more rigorous genetic approaches such as CRISPR. All the experiments are in cell lines of dubious physiological relevance (e.g., HeLa) rather than in primary cells (or in vivo). Thus, the physiological relevance of the findings is not entirely clear, and there is a chance that none of these observations are actually relevant to human shigellosis. There are very few experiments that demonstrate that the response affects bacterial replication, and none that address the possible in vivo inflammatory consequences during shigellosis disease. Although the authors model implies that Caspase-8 activation benefits the host, this is not actually demonstrated. Nevertheless, I tend to think that the paper is a significant contribution to the literature that advances the field. I think that the limitations should be explicitly acknowledged and addressed by a paragraph in the discussion.

Other points to be addressed:

1. Line 101: "The concept of effector-triggered immunity in eukaryotes was only recently proposed". This is incorrect. ETI was described in plants in the 1990s. Perhaps an acknowledgement of the plant literature would be appropriate here.
2. Line 128-130. This sentence ("We also demonstrate...") has grammatical issues.
3. Central to the authors model is the claim that infected cells are responding to TNF, but there is no discussion as to the source of the TNF, and no direct demonstration that TNF is being produced during the infection. This is especially interesting since TNF production would presumably be NF- κ B-dependent and should therefore be blocked by *OspI*. One might therefore expect more TNF to be induced by the *OspI* mutant, and yet, the effects of TNF that are described are in the presence of *OspI* (e.g., wild-type bacteria). This could be addressed by a TNF ELISA of supernatants from a simple timecourse of infection of HeLa and HT-29 cells with WT and *OspI* mutant *Shigella*.

4. How does Ospl-inhibition of Ubc13 result in cIAP degradation? This is central to the authors model but it is not explained how this occurs. The authors should discuss what they think is happening here.
5. Line 171: avoid phrases like "To prove our hypothesis" as this suggests the authors are approaching these experiments with a preconceived bias. Better would be "To test our hypothesis...". A similar issue exists on Line 277.
6. Line 172: which other Shigella deletion mutants were tested? Be specific and show the data or don't mention the other mutants.
7. Line 292: it is no longer true that it is unknown whether OspC1/C2/C3 have enzymatic activity. It is not reasonable for the authors to pretend they discovered this enzymatic activity. Delete "During our manuscript preparation" and simply acknowledge the prior literature. To be clear, the 2021 Li et al Nature paper did address the enzymatic activity of OspC1 in addition to OspC3.
8. Line 597: it is claimed OspD3 targets RIPK1 and RIPK3 via its protease activity. This is not shown in the manuscript so at a minimum needs a reference. As a note, I do not believe that the authors' previous paper on OspD3 (Ashida et al 2020) formally demonstrated that OspD3 is a protease. The only experiments shown are in cells, therefore it is possible OspD3 activates a protease rather than itself being a protease. A formal demonstration of protease activity would require purified proteins.
9. The authors previous work (Ashida 2020) showed OspD3 triggers the efficient degradation of RIPK1. However, the present work demonstrates that RIPK1 is important for Caspase-8 activation (Fig 1C shows Caspase-8 activation is blocked by the RIPK1 inhibitor Necrostatin-1). Can the authors explain how Nec-1 is able to have effects if RIPK1 is already degraded by OspD3?
10. There are some data in mouse models demonstrating that Caspase-8 and TNF restricts Shigella which might be discussed to support the in vivo relevance of the authors findings.
11. Other effectors that inhibit NF- κ B have been reported (e.g., OspG, IpaH1.4, OspZ). It is therefore surprising that loss of just Ospl has effects. The authors should discuss whether they think these other effectors also contribute, or explain why there isn't redundancy. I note that in Fig 2G there is some evidence for OspC1-inhibitable Caspase-8 activation by the Ospl mutant strain, suggesting that these other effectors may also contribute to Caspase-8 activation.
12. Figure 3B. Specify in the figure that control and Ubc13 are siRNAs
13. Figure 6 legend description for gentamicin protection assay is confusing and does not match the methods section. Was the infection allowed to proceed for 12 hours before gent was added as the legend implies or was it a 30 minute invasion followed by washes and 12 hour incubation with gent as the methods imply?

Referee #3:

This is a very thorough study assessing how caspase-8 is activated by Shigella in a TNF receptor plus RipK1 dependent fashion. An elegant set of experiments shows how ospC1 helps to prevent this caspase-8 activation in epithelial cells presumably via ADP-ribosylation. And the study identifies Ospl as one T3SS effector which triggers caspase-8 mediated apoptosis via Ubc13/cIAP inactivation.

The work provides a well-balanced set of in vitro approaches combining tissue culture infection models, stable cell lines, transfection, immunoblotting, siRNA knockdown, immunoprecipitation, treatment with ligands and/or inhibitors. I only have a few minor points which the authors may want to address to further strengthen their work.

1. The work nicely establishes effector-triggered immunity in cancer cell lines (HeLa or HT29). It would seem interesting to know, if this is also true for primary human epithelium cells. Such experiments should be feasible with the recent advances in organoid technologies and the authors should at least discuss what could be different in such non-transformed systems. Would Ospl and OspC1 control the ratio of epithelial cells that undergo cell death before expulsion into the gut lumen?

2. Shigella shows a striking host specificity for humans, while it cannot efficiently infect wild type mice. It would be interesting to discuss if any of the molecular interactions discovered in this work show species-specific differences which may offer an explanation for this interesting phenomenon.

3. Line 171-173: the authors describe a search for Shigella T3SS effectors that trigger caspase-8 activation and mention 'various effector gene deletion mutants', but the only effector mutant shown and described is ospl. Were other effector mutants actually tested in these experiments? It could be convincing to see other effector mutants do not show these phenotypes (if tested); otherwise this sentence could be reworded to reflect that only ospl was tested.

4. Materials and Methods indicate that HeLa cells were used. However, several Figure legends indicate that HT29 were employed. The authors may want to clarify which cells were used where.

5. I would suggest these comments be slightly more restrained; certainly the authors provide evidence for 'a' mechanism of effector-triggered immunity, but it is not necessarily the comprehensive mechanism of cell death and countermeasures - this manuscript rather seems an example of a set of 3 effectors that can manipulate/trigger different aspects of cell death signalling. There may well be other undescribed effectors that contribute to such phenotypes, so I would suggest more conservative phrasing.: Line 46-48: "we describe the mechanism of induction of cell death via effector-triggered immunity and the bacterial countermeasures" and; Line 411-412: "we describe the comprehensive mechanism of cell death induction known as effector-triggered immunity and the bacterial countermeasures during Shigella infection"

Referee #4:

In this manuscript by Ashida et al., the authors characterize the interplay between the Shigella OspI, OspC1 and Ospd3 type III effectors that divert and counteract cell death pathways. While the enzymic activity of these effectors was characterized in previous works, the current studies clarifies the targets of OspI and OspC1 and their respective role in regulating cell death. It is shown that in addition of inhibiting NF- κ B activation, OspI also inactivates Ubc13 to target cIAPs and trigger caspase 8 activation. OspC1 is shown here to inactivate caspase 8 by ADP-ribosylation, an activity that was inferred from recently published works on the related OspC2 and OspC3 effectors on other caspases. The subsequent inactivation of caspase 8 by OspC1 triggers necroptosis, this latter being inhibited by Ospd3.

This is an interesting and comprehensive piece of work that highlights the distinct and complementary roles of Shigella effectors in delaying cell death, a critical issue for successful tissue colonization. The experiments are generally convincing and support the conclusions. Despite the density and complexity of the system, the text is clear and well presented. I only have a few suggestions.

Major comments:

1. In Figs. 2B, 2E, 3G, 4 and 6D, caspase 8 activation is simply analyzed by Western blot visualization of its cleavage product and not quantified in Luminescence assays. In instances, the caspase 8 cleavage product is barely visible or appears to differ in intensity relative to controls. These experiments should be strengthened by densitometry analysis of the Western blot signals, the number of independent replica should be mentioned, and statistical testing performed.
2. The authors claim that OspI inactivates Ubc13, thereby impairing cIAP and inducing caspase 8 activation. Analysis of the effects of a RIPK mutant deficient for apoptosis in response to TNF- α but not TLR signaling such as the K627R in the RIPK death domain would be a good addition to show the role of OspI in TNFR signaling mediated caspase 8 activation.

Minor comments:

3. Fig. 1B. Several casp 8 cleavage products are observed in response to TNF- α inhibitor or in a T3SS-dependent manner during Shigella challenge. Can the authors comment / clarify?
4. Figs. 4A, B should be commented in the text and would probably better fitted to the EV data.

Referee #1:

In their manuscript, the authors have characterised the infection of Shigella strains in HT29 and HeLa cells in vitro. This is a severe limitation as it is not clear whether this also happens in primary tissues during native infection. Using a series of well-controlled experiments, the authors have investigated single effector mutants that block different steps of inter-connected programmed cell pathways. These include RIPK1-caspase-8-dependent apoptosis and RIPK1-RIPK3-dependent necroptosis. They use single mutants of ospC1 (a caspase-1,3,4,8 inhibitor), ospI (inhibitor of Ube13, a ubiquitin conjugating enzyme) and ospD3 (a protease that inactivates RIPK1 and RIPK3). Every effector they have studied here has been extensively characterised before, and the main novelty is looking at them together, but even that has been done previously by the same group (0.15252/embj.2020104469). Their experiments suggest that Shigella infection leads to interference at distinct steps that affect programmed cell death pathways. While it is nice to see the three effectors compared side-by-side, much of this is expected from previous work, and indeed discussed elsewhere based on the known blockades that trigger alternative signalling (e.g., caspase-8 blockade is required for necroptosis, NF- κ B blockade does influence apoptosis, among other effects). I also do not agree this is effector-triggered immunity; this merely shows that these effectors work together as part of a network, which is also the case with other Gram-negative enterics (e.g., Salmonella, EPEC/Citrobacter). Therefore, overall, the advance is limited. As far as I can tell, the main new findings are the specific effect of Ube13 on IAPs and OspC1 on caspase-8 modification and activity. It is also likely that their findings only apply to these specific cancer cell lines, which prevents generalisation of these findings to the actual human infection. In particular, HT29 is a Caspase-8 heterozygous line (one wildtype and a p.Q406 copy, which affects caspase-8 functions), and therefore has unique programme cell death features not seen in 'wildtype' homozygous cells (10.1158/0008-5472.CAN-14-0013; 10.1158/0008-5472.CAN-12-3342). HeLa cells (which are not intestinal in origin) do not express RIPK3 and therefore naturally do not undergo necroptosis; therefore, they are not suitable for studies on the interconnectedness between apoptosis and necroptosis. It is also not clear why some experiments are performed in HT29 and some in HeLa and no scientific rationale for this is provided (e.g., Why are several OspI experiments in HeLa?). Given that most cancer cells have unusual cell death signalling, much of this work needs to be carried out in primary intestinal cells (e.g., organoids) or more native systems to be convincing and generalisable to natural infection by Shigella. Other inconsistencies and lack of support for some of the conclusions detailed below:*

> We thank Referee #1 very much for their supportive comments on our manuscript.

(i) We respectfully disagree with Referee #1's opinion that our manuscript is lacking in terms of novelty and contributions. In the arms race between bacterial pathogens and host defense systems, both pathogens and hosts deploy offensive and defensive strategies for their own benefit. At this point, we believe our findings show an important interconnection between bacterial effector function and host cell death induction, which we believe to be quite novel. As noted by Referee #1, previous work identified NF- κ B blockade does influence apoptosis, but our finding further revealed that inactivation of Ubc13, but not blockade of NF- κ B by the OspI effector, could lead to caspase-8 activation (please see Comment No. 11 to Referee #2). In addition to our findings, Yeap et al. recently revealed that the complex interplay between NleE, NleB, and EspL effector of *Citrobacter rodentium* enables them to subvert innate immune signaling, apoptosis, necroptosis, and NLRP3 activation, which has been published in *EMBO J* (Yeap et al. 2025; PMID: 40128366). These findings allow us to piece together previously established findings in the field into an interesting overall model that we believe makes an important contribution to our understanding of the host response to bacterial pathogens, including *Shigella*. We expect these findings to become even more critical in the future.

(ii) We agree that HT-29 is a Caspase-8 heterozygous cell line; however, many researchers have used it as a tool for studying necroptosis (e.g., Zhou and Yuan. *Cell*. 2014; He et al. *J. Exp. Med.* 2017; Pearson et al. *Nat. Microbiol.* 2017; Hefele et al. *Mucosal Immunol.* 2018; Shubina et al. *J. Exp. Med.* 2020). Our data have shown that HT-29 cells responded correctly to stimulants (TNF- α) and inhibitors (BV-6 or CHX) and appropriately induced apoptosis or necroptosis (revised Fig. EV6A and EV6B). Therefore, we believe that HT-29 is an appropriate cell line for investigating necroptosis. However, in response to Referee #1's concern, we have provided additional data using other intestinal epithelial cell lines, including HCT116, T84 and HT-55, which showed the same results as HT-29 cells, in revised Fig. EV2, EV5, and EV6. These data strongly support our conclusion that Ubc13 inactivation by OspI is required for caspase-8–apoptosis and that the blockade of caspase-8 by OspC1 is required for necroptosis. To further address Referee #1's concern, we added a paragraph to the Discussion regarding the limitations of this study with regard to *in vivo* model or primary cells (Lines 495–505).

(iii) We regret that our explanation in the text lacked detail. As pointed out by Referee #1, we used HeLa cells instead of HT-29 cells in the experiment shown in the original Figure 3 because of the low knockdown efficiency of *Ubc13*-targeting siRNAs (please see Response Fig. 1). However, we found the knockdown efficiency of *Ubc13*-targeting siRNAs was sufficient in HeLa (HCT116) cells (Response Fig. 1 and revised Fig. EV3). Therefore, we selected HeLa cells for *Ubc13* knockdown experiments in the original Fig. 3. Indeed, HeLa cells are widely used as a model to investigate the interaction between *Shigella* and host cells even in recent years (e. g., Li et al. Nature 2021; Luchetti et al. Cell Host Microbe. 2021; Alphonse et al. Cell 2022). We also added a description of knockdown efficiency (Lines 254–256).

Although we have confirmed that HT-29 cells stably expressing Ospl (HT-29/ospl) exhibited caspase-8 activation under TNF- α stimulation (please see Response Fig. 2), we have provided HeLa/ospl data to be consistent with cells used in *Ubc13* knockdown experiments in Fig. 3.

Knockdown efficiency of *Ubc13* in HT-29 or HeLa cells

HT-29 or HeLa cells were treated with indicated siRNAs.
Cell lysates were subjected to immunoblotting.

Response Fig. 1 to Referee

HT-29, HT-29/ospl, or HT-29/ospl-CA cells treated with or without TNF- α (25 ng/ml) were incubated for 24 h. Cell lysates were subjected to immunoblotting.

Response Fig. 2 to Referee

We have taken all of these comments into account and resubmitted a revised version of our manuscript. The following are our point-by-point replies to Referee #1's comments.

Major points.

1. Unless these findings are shown in more native systems, much of the work is not novel and a repeat of what has previously been shown with individual effectors or families of effectors. Much of the work in HT29 should be repeated at least in an intestinal epithelial cell line (there are several - refer to the CRC database <http://www.coloncanceratlas.org/>) that has no mutations (at least) in the CASP8 gene and others in the pathways being addressed.

> Thank you for your supportive comment. To address this, we have repeated the experiments in other intestinal epithelial cell lines, such as HCT116, T84, and HT-55, which has no mutations

in the *CASP8* gene. We provided additional data using these other intestinal epithelial cell lines, which showed the same results of HT-29 cells, in revised Fig. EV2B, EV2C, EV3B, EV3E, EV5A–C, and EV6C. These data strongly support our conclusion that Ubc13 inactivation by Ospl is required for caspase-8 apoptosis and that caspase-8 blockade by OspC1 is required for necroptosis.

In addition, we followed the suggestions of Referees #2 and #3 (please see our comment to Referee #2 and Comment No. 1 to Referee #3) and added a paragraph in the Discussion regarding *in vivo* model or primary cells (Lines 495–505).

2. To claim TNFR1 is driving death, did these cells produce TNF during infection - either in the supernatant or membrane-bound? Have the authors tested anti-TNF antibodies instead of a small molecule inhibitor, which can have off-target effects?

> To address this comment, we performed ELISA assays in *Shigella*-infected cells. We have confirmed that these cells produced TNF- α , which was released into the supernatant during infection. We have provided additional data and information in the revised Fig. 1D as well as in the Material & Methods section (see the ELISA section). In addition, we performed anti-TNF antibody treatment to test the importance of TNFR1 in caspase-8 activation in the revised Fig. 1C and confirmed that anti-TNF antibody prevented caspase-8 cleavage in Δ *ospC1*-infected cells. These data strongly indicate that TNFR1 signaling is required for caspase-8 activation during *Shigella* infection.

3. There are inconsistencies as well - why is TNF-dependent IL8 production not affected by ospl in Fig EV1? This is not in line with findings elsewhere in the paper.

> Previous studies have indicated that Ubc13 is not required for TNF- α -stimulated NF- κ B activation (Xu et al. Mol. Cell 2009; Tokunaga et al. Nat. Cell Biol 2009). Our data also show that *Ubc13* knockdown suppressed PMA-stimulated, but not TNF- α -stimulated, NF- κ B-regulated *IL8* expression (revised Fig. EV3C). Consistent with *Ubc13* knockdown data, Ospl suppressed PMA-stimulated, but not TNF- α -stimulated, expression of the NF- κ B-regulated gene *IL8* (revised Fig. EV1D). Therefore, inactivation of Ubc13 by Ospl selectively inhibits PMA-stimulated, but not TNF- α -stimulated, IL-8 production.

In contrast, our data show that *Ubc13* knockdown increased caspase-8, caspase-3, and PARP activation, which are hallmarks of apoptosis, in response to TNF- α stimulation. However, this did not occur in response to IL-1 β or PMA (Fig. 3A, 3B, EV3A, and EV3B). Consistent with *Ubc13* knockdown data, *Ospl* triggers caspase-8, caspase-3, and PARP activation under TNF- α stimulation, but not IL-1 β or PMA (revised Fig. Fig. 2B, Fig. EV1A and EV1B). Therefore, we concluded that inactivation of *Ubc13* by *Ospl* affects the TNF- α –caspase-8 pathway but not the TNF- α –NF- κ B pathway. These data strongly indicate that *Ospl* triggers caspase-8–apoptosis activation in an NF- κ B-independent manner.

4. To convincingly show TRADD/FADD-independent apoptosis, it would be good to also show whether the silencing did reduce TNF or Fas-dependent apoptosis in these settings. Without this it is plausible that low levels of these adaptors despite RNAi suffice for cell death.

> We have confirmed that the knockdown efficiency of TRADD and FADD siRNA was sufficient (revised Fig. 1H inset). Furthermore, we have investigated these siRNAs-mediated knockdown of TRADD and FADD prevented TNF- α + CHX dependent casase-8 activation (please see Response Fig. 3). We believe these data further support our conclusion.

HT-29 cells treated with control, TRADD, or FADD siRNAs were treated with TNF- α (25 ng/ml) + BV-6 (1 μ M) or TNF- α (25 ng/ml) + CHX (25 μ g/ml) were incubated for 8 h. Cell lysates were subjected to immunoblotting.

Response Fig. 3 to Referee

5. The authors test various mutants of *Shigella*, but do not demonstrate that the level of infection is comparable across strains. This is an important control experiment.

> Growth in culture medium and invasion of HT-29 and HeLa cells was determined for all *Shigella* mutant strains prior to use in this study (please see Response Fig. 4). We have confirmed the level of infection was comparable across strains.

(A-D) HT-29 (A-C) or HeLa (D) cells were infected with the indicated *Shigella* strains. 30 min (A-C) or 15 min (D) later, cells were washed and treated with gentamicin for an additional 15 min. After incubation, the cells were washed three times with PBS, and then lysed with PBS containing 0.5% Triton X-100. The lysates were diluted with PBS and plated onto LB-agar plates, and the number of intracellular bacteria was determined by calculating CFU (colony-forming units).

Response Fig. 4 to Referee

6. It would be informative to compare relative activity of Cas8 during infection conditions and TNF treatments +/- RIPK1 inhibitor.

> To address this comment, we provided the data showed in the original Fig.1D for both caspase-8 activity (RLU) and relative caspase-8 activity (%) as Response Fig. 5. As shown in Response Fig. 5, the same result was observed for both caspase-8 activity (RLU) and relative caspase-8 activity (%). However, because we believe it would be more informative to compare RLU than the relative activity of caspase-8 (%), we would like to keep our original Figure.

HT-29 cells infected with the indicated *Shigella* strains or treated with TNF- α (25 ng/ml) + BV-6 (1 μ M), or TNF- α (25 ng/ml) + CHX (25 μ g/ml) in the presence or absence of Nec-1 (50 μ M) were incubated for 6 h. Cell lysates were subjected to measurement of caspase-8 activity.

(left) Caspase-8 activity is reported as relative light units (RLU) of infected samples normalized to the value in uninfected samples.

(right) Caspase-8 activity is reported as relative percentage to the each value of Nec-1 nontreated samples (Nec-1 (+)/Nec-1 (-)).

Data are expressed as the mean \pm SD from triplicate wells.

*P < 0.05; n.s., not significant (unpaired two-tailed Student's t-test).

Response Fig. 5 to Referee

Minor points

1. The authors use 'loose' language in several places (e.g., line 193, 204), such as "ospl induces caspase-8...", this is not accurate as it implies a direct action. We know it is very indirect and the authors should only state an "ospl mutant leads to ..." or "loss of ospl results in..."

> In accordance with this suggestion, we have corrected this sentence in the revised text (Lines 216 and 227).

2. On line 171 - do not say "prove this hypothesis", hypothesis should only be tested and results reported.

> In accordance with this suggestion, we have corrected this sentence in the revised text (Line 192).

Referee #2:

This interesting paper investigates how Shigella is sensed by the innate immune system. Shigella is a very important bacterial pathogen that remains poorly understood. The question of how Shigella is sensed by the innate immune system is an important question to address. The authors propose Shigella is sensed via an "effector-triggered" immune response. This response is induced by a secreted Shigella effector called OspI that inhibits NF-κB signaling. Inhibition of NF-κB signaling presumably evolved to benefit the bacteria, but in response, the host has evolved to sense that its NF-κB pathway has been disrupted and respond by activating a Caspase-8-dependent cell death response. Interestingly, another Shigella effector, OspC1, has evolved to inhibit Caspase-8. Thus there is a fascinating attack and counterattack (or "arms race") between Shigella and its human hosts. Many of the key aspects of this paper were previously known. For example, it was known that (a) OspI inhibits NF-κB, (b) inhibition of NF-κB signaling alters TNF signaling from a pro-survival to a pro-death output, (c) OspC1 was already shown to block Caspase-8. The main novel contributions of the manuscript are to show that Caspase-8 activation induced by Shigella depends on inhibition of NF-κB by OspI, and that the OspI-induced Caspase-8 activation also depends on signaling via the TNF receptor. These observations allow the authors to piece together previously established findings in the field into an interesting overall model that is an important contribution to our understanding of the host response to Shigella. The main limitation of the paper is that the experimental methods are not state-of-the-art. There is a lot of reliance on inhibitors and siRNAs (often just a single siRNA) instead of more rigorous genetic approaches such as CRISPR. All the experiments are in cell lines of dubious physiological relevance (e.g., HeLa) rather than in primary cells (or in vivo). Thus, the physiological relevance of the findings is not entirely clear, and there is a chance that none of these observations are actually relevant to human shigellosis. There are very few experiments that demonstrate that the response affects bacterial replication, and none that

address the possible in vivo inflammatory consequences during shigellosis disease. Although the authors model implies that Caspase-8 activation benefits the host, this is not actually demonstrated. Nevertheless, I tend to think that the paper is a significant contribution to the literature that advances the field. I think that the limitations should be explicitly acknowledged and addressed by a paragraph in the discussion.

> We thank Referee #2 for these supportive comments on our manuscript.

(i) We have tried several times to generate Ubc13 knockout cell lines (HT-29, HeLa, or HCT116) using the CRISPR/Cas9 system (TrueCut™ Cas9 Proteins System [Invitrogen]). We prepared four sets of synthetic guide RNAs against human Ubc13 with the following sequences: 5'-GGCGUUGCUCUCAUCUGGUU-3', 5'-CCCUCAAAGGGGAAUCCUG-3', 5'-CCAUUGGGUAAUUCUUCUGGA-3', 5'-CAUCCUAAUGUAGACAAGUU-3'. However, because we could not obtain *Ubc13* knockout cell lines, we selected *Ubc13* siRNA knockdown experiments to show the importance of the inactivation of Ubc13 in caspase-8 activation. We were able to obtain cIAP1/2-double knockout cells. We have provided additional data in revised Fig. 4F. In siRNA knockdown experiments, we used three siRNAs targeting *Ubc13*. In response to Referee #2's concern, we have provided additional data and information in the revised Figure EV3, as well as in the Materials & Methods section (please see RNAi section). Additionally, please see Comment (i)–(iii) and Response Fig. 1 and 2 to Referee #1.

(ii) According to Referee #2's suggestion, we have added a paragraph to the discussion regarding the limitations of this study in terms of the *in vivo* model or primary cells (Lines 495–505). In addition, to address Referee #2's concern regarding whether caspase-8 activation is beneficial for the host, we provided additional data that OspI-mediated caspase-8 activation reduces the intracellular bacterial number (revised Fig. 2I).

We have taken all of these comments into account and resubmitted a revised version of our paper. The following are our point-by-point replies to Referee #2's comments.

Other points to be addressed:

1. Line 101: "The concept of effector-triggered immunity in eukaryotes was only recently proposed". This is incorrect. ETI was described in plants in the 1990s. Perhaps an acknowledgement of the plant literature would be appropriate here.

> As suggested, we have corrected this sentence in the revised text (Lines 113–115).

2. Line 128-130. This sentence ("We also demonstrate...") has grammatical issues.

> In line with this suggestion, we have now corrected this sentence in the revised text (Lines 143–145).

*3. Central to the authors model is the claim that infected cells are responding to TNF, but there is no discussion as to the source of the TNF, and no direct demonstration that TNF is being produced during the infection. This is especially interesting since TNF production would presumably be NF- κ B-dependent and should therefore be blocked by *OspI*. One might therefore expect more TNF to be induced by the *OspI* mutant, and yet, the effects of TNF that are described are in the presence of *OspI* (e.g., wild-type bacteria). This could be addressed by a TNF ELISA of supernatants from a simple timecourse of infection of HeLa and HT-29 cells with WT and *OspI* mutant *Shigella*.*

> As suggested, we have performed an ELISA-based time course analysis of TNF- α production in supernatants from HT-29 cells infected with *Shigella* WT and mutant strains. As shown in revised Fig. 1D and Fig. EV2F, TNF- α was being produced during the infection, suggesting that the secreted TNF- α can activate TNFR–caspase-8–apoptosis signaling. However, although a higher amount of TNF- α was produced in Δ *ospI* than with WT infection at early time points (2 h post infection), there was no significant difference in TNF- α production between WT and Δ *ospI* during later infections (4–8 h post-infection). In contrast, IL-8 production was significantly higher in Δ *ospI*-infected cells than it was in WT (2–8 h post-infection; revised Fig. EV2F).

Because the time course study of caspase-8 activation in Δ *ospC1*-infected cells revealed that the caspase-8 stimulus began to accumulate after 5 h post-infection, there was no significant difference in TNF- α production between WT and Δ *ospI* in terms of the time when caspase-8 became activated (please see Comment No. 9 to Referee #2). Therefore, we concluded that, while higher amounts of TNF- α were produced in Δ *ospI* than with WT infection at early time points, this difference didn't affect caspase-8 activation levels.

4. How does OspI-inhibition of Ubc13 result in cIAP degradation? This is central to the authors model but it is not explained how this occurs. The authors should discuss what they think is happening here.

> To address this comment, we added some additional discussion to the text (Lines 461–471). However, because we are currently preparing a separate manuscript regarding the molecular mechanism of cIAP's inhibition by OspI, we have chosen not to include the data regarding the mechanism of cIAP degradation in this manuscript.

5. Line 171: avoid phrases like "To prove our hypothesis" as this suggests the authors are approaching these experiments with a preconceived bias. Better would be "To test our hypothesis...". A similar issue exists on Line 277.

> In accordance with this suggestion, we corrected this sentence in the revised text (Line 192).

6. Line 172: which other Shigella deletion mutants were tested? Be specific and show the data or don't mention the other mutants.

> We have performed a caspase-8 activity assay using several *Shigella* effector gene deletion mutants. We have now added the data regarding Caspase-8 activity with infections by *Shigella* deletion mutants to Appendix Fig. 1.

7. Line 292: it is no longer true that it is unknown whether OspC1/C2/C3 have enzymatic activity. It is not reasonable for the authors to pretend they discovered this enzymatic activity. Delete "During our manuscript preparation" and simply acknowledge the prior literature. To be clear, the 2021 Li et al Nature paper did address the enzymatic activity of OspC1 in addition to OspC3.

> We agree and have corrected the text (Lines 329–332) accordingly.

8. Line 597: it is claimed OspD3 targets RIPK1 and RIPK3 via its protease activity. This is not shown in the manuscript so at a minimum needs a reference. As a note, I do not believe that the authors' previous paper on OspD3 (Ashida et al 2020) formally demonstrated that OspD3 is a

protease. The only experiments shown are in cells, therefore it is possible Ospd3 activates a protease rather than itself being a protease. A formal demonstration of protease activity would require purified proteins.

> We agree with Referee #2's comment. However, because EspL, the Ospd3 homolog of EPEC, has already been confirmed to be a cysteine protease using purified proteins, we believe Ospd3 is also a protease. To address this comment, we have corrected the sentence and added reference (Pearson et al. 2017 Nat. Microbiol.) in the revised text (Lines 133–134).

9. The authors previous work (Ashida 2020) showed Ospd3 triggers the efficient degradation of RIPK1. However, the present work demonstrates that RIPK1 is important for Caspase-8 activation (Fig 1C shows Caspase-8 activation is blocked by the RIPK1 inhibitor Necrostatin-1). Can the authors explain how Nec-1 is able to have effects if RIPK1 is already degraded by Ospd3?

> We appreciate this important comment. As pointed out by Referee #2, Ospd3 cleaves RIPK1. We have previously performed a cell-death time-course experiment. As shown in Response Fig. 6 (Ashida et al. 2020. Fig. 5C), the time-course study of caspase-8 activation in $\Delta ospC1$ -infected cells revealed that the caspase-8 stimulus began to accumulate 5 h post-infection. Accordingly, the levels of MLKL phosphorylation in $\Delta ospD3$ -infected cells began to increase after the 6 h time point. These data support that Ospd3 begins to degrade RIPK1 after 6 h. Therefore, we believe that RIPK1-dependent caspase-8 activation was elevated in *Shigella*-infected cells before Ospd3 degraded RIPK1.

HT-29 cells were infected with *Shigella* WT, $\Delta ospC1$, or $\Delta ospD3$ strains. Cell lysates obtained at the indicated time points were subjected to immunoblotting. (Ashida et al. 2020. EMBO J; PMID: 32657447. Fig. 5C)

Response Fig. 6 to Referee

10. There are some data in mouse models demonstrating that Caspase-8 and TNF restricts *Shigella* which might be discussed to support the *in vivo* relevance of the authors findings.

> As suggested, we have added a description of Caspase-8 and TNF- α in the revised text (Lines 151–152 and 164–166).

11. Other effectors that inhibit NF- κ B have been reported (e.g., *OspG*, *IpaH1.4*, *OspZ*). It is therefore surprising that loss of just *OspI* has effects. The authors should discuss whether they think these other effectors also contribute, or explain why there isn't redundancy. I note that in Fig 2G there is some evidence for *OspC1*-inhibitable Caspase-8 activation by the *OspI* mutant strain, suggesting that these other effectors may also contribute to Caspase-8 activation.

> We believe that *OspI* triggers caspase-8 activation via Ubc13 inactivation rather than by NF- κ B inhibition. To support this idea, we showed that BV-6 treatment rescued caspase-8 activation in cells infected with $\Delta ospC1\Delta ospI$, but not TPCA-1 (IKK- β inhibitor) or BAY11-7082 (NF- κ B inhibitor) treatments (revised Fig. 4A, EV4A, EV4B and EV4C). Additionally, please see Comment No. 3 to Referee #1.

To address Referee #2's suggestion, we selected the $\Delta ospG$ mutant because our YSH6000 strain lacks the expression of *ipaH1.4* and *ospZ*. We infected the *Shigella* WT and $\Delta ospG$ strain in HT-29 cells and measured caspase-8 activity. In our results, deletion of *ospG* had no effect on

caspase-8 activity (revised Fig. EV4D). Furthermore, infection with $\Delta ospC1\Delta ospG$ still induced cleaved caspase-8 expression whereas $\Delta ospC1\Delta ospI$ did not lead to caspase-8 activation (Fig. EV4E and Fig. 2G). Identifying the involvement of NF- κ B inhibition by *Shigella* effectors in caspase-8 activation will be an important priority for future studies.

12. Figure 3B. Specify in the figure that control and Ubc13 are siRNAs

> In accordance with this suggestion, we have corrected Figure 3B.

13. Figure 6 legend description for gentamicin protection assay is confusing and does not match the methods section. Was the infection allowed to proceed for 12 hours before gent was added as the legend implies or was it a 30 minute invasion followed by washes and 12 hour incubation with gent as the methods imply?

> We regret the lack of detailed explanation in the legend. As suggested, we corrected this sentence in the revised Figure 6 legend (Lines 652–661). We have also corrected the Materials and Methods section (Lines 833–841).

Referee #3:

This is a very thorough study assessing how caspase-8 is activated by Shigella in a TNF receptor plus RipK1 dependent fashion. An elegant set of experiments shows how ospC1 helps to prevent this caspase-8 activation in epithelial cells presumably via ADP-ribosylation. And the study identifies OspI as one T3SS effector which triggers caspase-8 mediated apoptosis via Ubc13/cIAP inactivation. The work provides a well-balanced set of in vitro approaches combining tissue culture infection models, stable cell lines, transfection, immunoblotting, siRNA knockdown, immunoprecipitation, treatment with ligands and/or inhibitors. I only have a few minor points which the authors may want to address to further strengthen their work.

> We thank Referee #3 for these supportive comments on our manuscript. We have taken all of these comments into account and resubmitted a revised version of our paper.

The following are our point-by-point replies to their comments.

1. The work nicely establishes effector-triggered immunity in cancer cell lines (HeLa or HT29). It would seem interesting to know, if this is also true for primary human epithelium cells. Such experiments should be feasible with the recent advances in organoid technologies and the authors should at least discuss what could be different in such non-transformed systems. Would OspI and OspC1 control the ratio of epithelial cells that undergo cell death before expulsion into the gut lumen?

> In accordance with this suggestion, we have added discussion regarding the limitations of this study (Lines 495–505).

2. Shigella shows a striking host specificity for humans, while it cannot efficiently infect wild type mice. It would be interesting to discuss if any of the molecular interactions discovered in this work show species-specific differences which may offer an explanation for this interesting phenomenon.

> As suggested, we have added data and discussion regarding the species-specificity of the OspC1 effector (revised Fig. 5C and Lines 350–354).

Interestingly, our data showed that OspC1 catalyzed ADP-ribosylation of human caspase-8 but not of mouse caspase-8. As previously suggested by Luchetti et al. (Cell Host Microbe 2021; PMID: 34492225), this species-specific difference of the OspC1 effector might be an alternative explanation for *Shigella*'s specificity toward humans. We plan to address this possibility in future work.

3. Line 171-173: the authors describe a search for Shigella T3SS effectors that trigger caspase-8 activation and mention 'various effector gene deletion mutants', but the only effector mutant shown and described is ospI. Were other effector mutants actually tested in these experiments? It could be convincing to see other effector mutants do not show these phenotypes (if tested); otherwise this sentence could be reworded to reflect that only ospI was tested.

> We have performed a caspase-8 activity assay using several *Shigella* effector gene deletion mutants. As suggested, we added the data regarding Caspase-8 activity following infection by *Shigella* deletion mutants in Appendix Fig. 1.

4. *Materials and Methods indicate that HeLa cells were used. However, several Figure legends indicate that HT29 were employed. The authors may want to clarify which cells were used where.*

> In response to this comment, we have now corrected the Figure legends to clarify which cells were used in each case.

5. *I would suggest these comments be slightly more restrained; certainly the authors provide evidence for 'a' mechanism of effector-triggered immunity, but it is not necessarily the comprehensive mechanism of cell death and countermeasures - this manuscript rather seems an example of a set of 3 effectors that can manipulate/trigger different aspects of cell death signalling. There may well be other undescribed effectors that contribute to such phenotypes, so I would suggest more conservative phrasing.: Line 46-48: "we describe the mechanism of induction of cell death via effector-triggered immunity and the bacterial countermeasures" and; Line 411-412: "we describe the comprehensive mechanism of cell death induction known as effector-triggered immunity and the bacterial countermeasures during Shigella infection"*

> As suggested, we have corrected these sentences in the revised text (Lines 60 and 451). In addition, we rephrased the title to "*Shigella* type III secretion system effectors fine-tune host inflammation and cell death".

Referee #4:

In this manuscript by Ashida et al., the authors characterize the interplay between the Shigella OspI, OspC1 and OspD3 type III effectors that divert and counteract cell death pathways. While the enzymic activity of these effectors was characterized in previous works, the current studies clarifies the targets of OspI and OspC1 and their respective role in regulating cell death. It is shown that in addition of inhibiting NF-κB activation, OspI also inactivates Ubc13 to target cIAPs and trigger caspase 8 activation. OspC1 is shown here to inactivate caspase 8 by ADP-ribosylation, an activity that was inferred from recently published works on the related OspC2 and OspC3 effectors on other caspases. The subsequent inactivation of caspase 8 by OspC1 triggers necroptosis, this latter being inhibited by OspD3. This is an interesting and comprehensive piece of work that highlights the distinct and complementary roles of Shigella

effectors in delaying cell death, a critical issue for successful tissue colonization. The experiments are generally convincing and support the conclusions. Despite the density and complexity of the system, the text is clear and well presented. I only have a few suggestions.

> Thank you very much for your supportive comments on our manuscript. We have taken all of these comments into account and resubmitted a revised version of our paper.

The following are our point-by-point replies to Referee #4's comments.

Major comments:

1. In Figs. 2B, 2E, 3G, 4 and 6D , caspase 8 activation is simply analyzed by Western blot visualization of its cleavage product and not quantified in Luminescence assays. In instances, the caspase 8 cleavage product is barely visible or appears to differ in intensity relative to controls. These experiments should be strengthened by densitometry analysis of the Western blot signals, the number of independent replica should be mentioned, and statistical testing performed.

> As suggested, we have now added the data regarding Caspase-8 activity (luminescence assay) to strengthen and improve our result in revised Fig. 2F, 3G, 3I, EV1A, and EV4B.

2. The authors claim that Ospl inactivates Ubc13, thereby impairing cIAP and inducing caspase 8 activation. Analysis of the effects of a RIPK mutant deficient for apoptosis in response to TNF- α but not TLR signaling such as the K627R in the RIPK death domain would be a good addition to show the role of Ospl in TNFR signaling mediated caspase 8 activation.

> Thank you for your supportive comment. However, because we are currently preparing a separate manuscript regarding the molecular mechanism of the cIAP inhibition by Ospl, we plan to investigate and elucidate this issue further in future studies.

Minor comments:

3. Fig. 1B. Several casp 8 cleavage products are observed in response to TNF- α inhibitor or in a T3SS-dependent manner during Shigella challenge. Can the authors comment / clarify?

> To address this comment, we have repeated this experiment. As pointed out by Referee #4, treatment using a higher concentration of the TNF- α inhibitor (5 μ M; lane 3) but not a lower concentration (1 μ M; lane 2) induced the appearance of a 40 kDa band using an anti-casp8 antibody, even when the cells were not stimulated (revised Fig. 1B). However, we couldn't detect cleaved caspase-8 (p18) band after the TNF- α inhibitor treatment (lane 2 and 3). Therefore, we believe that the "Several casp8 cleavage products" pointed out by Referee #4 are a non-specific band induced by the TNF- α inhibitor treatment.

4. Figs. 4A , B should be commented in the text and would probably better fitted to the EV data.

> We appreciate this supportive comment. However, the data regarding inhibitor treatment provided us clues suggesting that we should focus on cIAP as a key factor in OspI-mediated caspase-8 activation. We believe this Figure is important for our manuscript; therefore, we would like to retain Fig. 4A and B in the revised Figure rather than EV data.

Dear Dr. Ashida,

Thank you for submitting your revised manuscript (EMBOJ-2024-119594R) to The EMBO Journal for our consideration, and for your patience during peer review. Your manuscript has been sent back to three of the original referees that had previously reviewed the initial version of your manuscript, and we have now received their comments, which you can find below.

I am very pleased to say that all three referees are satisfied with the revision, mentioning that almost all comments raised in the previous round of review have been successfully addressed. The referees acknowledge that the new version of the manuscript is significantly improved.

However, referee #1 points out that there are remaining issues regarding the statistical analyses throughout the manuscript - which are also in line with issues raised by our data editors (please see their requests below). We totally agree with the referee that:

1. No statistics should be shown in the Figures for technical replicates; the shown statistics should always refer to average values (of technical replicates) across independent biological repeats.
2. Biological repeats should be shown in graphs throughout the manuscript, not representative graphs of technical replicates.
3. Please make sure you use the most suitable statistical test for each comparison, e.g. Student's t-test vs. ANOVA.
4. Information on exact p-values, the identity of the used statistical tests, and the definition of error bars are missing in several Figures; please see a detailed list provided by our data editors below.

I should note that all these issues must be fully addressed and the statistics corrected before we can accept the manuscript for publication in The EMBO Journal. Please submit a revised version of your manuscript with all textual changes highlighted (and revised Figures, if necessary) along with a detailed list of changes/corrections/explanations per Figure panel.

There are also a few other changes and corrections we need you to make in the final version of your manuscript:

- Please note that the reference format is not correct: for each reference, only the names of the first 10 co-authors should be listed, followed by "et al.". Please see our guide to authors for more information and revise your reference format as necessary: <https://www.embopress.org/page/journal/14602075/authorguide#referencesformat>.

- Please change the heading of your conflict-of-interest statement to "Disclosure and competing interests statement".

- The author contributions statement should be removed from the manuscript file. Instead, we use CRediT to specify the contributions of each author in the journal submission system. Please feel free to use the free text box to provide more detailed descriptions during submission. See also our guide to authors for more information: <https://www.embopress.org/page/journal/14602075/authorguide#authorshippinguidelines>.

- Main and Expanded View (EV) Figures must be uploaded as individual, high-resolution Figure files.

- The Appendix needs to be in PDF format; the Appendix Figure should be included in the Appendix PDF with its legend below the figure; the title page of the Appendix should contain the heading "Appendix for:" followed by the manuscript's title, and a Table of Contents with the page numbers for the listed items.

- Please note that EMBO press papers are accompanied online by:

- A) a short (2 sentences) summary of the findings and their significance,
- B) 2-5 short bullet points highlighting the key results, and
- C) a synopsis image in .jpg or .png format that is exactly 550 pixels wide and 300-600 pixels high (the height is variable). Please note that all text in the image needs to be legible at the final size.

Please upload this information along with your revised manuscript (the text for A and B should be provided in a separate Word file).

- During our routine data checks, our data editors have raised the following queries regarding figures, data, and legends. Please make sure that all requests below are completely addressed in the final version of your manuscript (please highlight all changes in the revised manuscript):

1. Please provide the exact p values in the legends of Figures 1F, 2A, C, D, F, H, I; 3B, D, E, G, I; 5E, 6A, C, E, G, H; EV1 D, EV2 F, EV3 C, EV6 B, S1.
2. Please indicate the statistical test used for data analysis in the legend of Figure 1D.
3. Please note that the error bars are not defined in the legends of Figures 1D, S1.

- The order of the manuscript sections must be corrected as follows: Title page - Abstract and Keywords - Introduction - Results - Discussion - Methods - Data Availability - Acknowledgements - Disclosure and Competing Interests Statement - References - Figure Legends - main Tables (if there are any) - Expanded View Figure Legends.

Please also note that as part of the EMBO publications' Transparent Editorial Process, The EMBO Journal publishes online a Peer Review File along with each accepted manuscript. This File will be published in conjunction with your paper and will include the referee reports, your point-by-point response and all pertinent correspondence relating to the manuscript. You can opt out of this by letting the editorial office know (contact@embojournal.org). If you do opt out, the Peer Review File link will point to the following statement: "No Peer Review File is available with this article, as the authors have chosen not to make the review process public in this case."

We look forward to seeing a final version of your manuscript as soon as possible. Please let us know if you have any questions and use this link to submit your revision: <https://emboj.msubmit.net/cgi-bin/main.plex>.

Best regards,

Ioannis

Referee #1:

In their revised manuscript, the authors have addressed almost all of the comments raised in review 1. The manuscript is now much improved as a result. I did, however, notice that statistics have been incorrectly applied in many instances or not fully described and these should be fixed. I do apologise I did not notice this in the first submission.

Examples below, but this applies throughout for graphs

1. In cases where triplicate wells are shown from one experiment, no statistics should be shown because it is meaningless to apply statistics to technical replicates. I hope statistics were applied to average values across biologically independent repeats. Ideally those repeats should be shown in graphs throughout (not representative graphs of technical replicates). For whatever reason if that is not possible, then graphs should be called representative and statistical analysis should not be shown.
2. In cases where there are 2 factors, e.g., Fig 1D, 1F, 2D, 3B, 3E, 3I etc (and similar ones elsewhere) - the authors have used Student's t tests, which is not correct. These data (averages from independent repeats, not technical repeats) should be analysed as 2-way ANOVAs. Student's t-tests are only appropriate when there are exactly 2 groups in the entire experiment.
3. It is not clear what stats were used for 1D, and some others, please add details.

Referee #2:

I am satisfied by the authors' response to my previous review.

Referee #4:

This is a significantly improved version that addresses my previous points.

Referee #1:

In their revised manuscript, the authors have addressed almost all of the comments raised in review 1. The manuscript is now much improved as a result. I did, however, notice that statistics have been incorrectly applied in many instances or not fully described and these should be fixed. I do apologise I did not notice this in the first submission. Examples below, but this applies throughout for graphs

> We thank the referee for these supportive comments on our manuscript. We have taken all of these comments into account and resubmitted a revised version of our paper.

1. In cases where triplicate wells are shown from one experiment, no statistics should be shown because it is meaningless to apply statistics to technical replicates. I hope statistics were applied to average values across biologically independent repeats. Ideally those repeats should be shown in graphs throughout (not representative graphs of technical replicates). For whatever reason if that is not possible, then graphs should be called representative and statistical analysis should not be shown.

> We apologize for causing a misunderstanding regarding the statistical analysis. Although Referee #1 claimed that no statistics should be shown in the Figures for technical replicates, we believe that our graphs are data from biological replicates, not from technical replicates, throughout the manuscript. We referred to papers by other groups (Broz et al. 2012 Nature [PMID: 22895188]; Kayagaki N et al. 2015 Nature [PMID: 26375259]; Peng et al. 2022 Mol Cell [35338844]).

For instance (Fig. 2A and Response Fig. 1), HT-29 cells were seeded in 96-well plate, and infected with *Shigella* WT, S325, $\Delta ospC1$ or $\Delta ospI$ at moi=10. Each strain was infected separately in three wells (performed in triplicate). After 8 h post infection, each infected cell lysates were subjected to measurement of Caspase-8 activity. Since all processes were performed independently in each well, we believe that our data are biological replicates rather than technical replicates. We repeated three times (three independent experiments) at same experimental condition, and confirmed the same conclusions in each experiments (Response Fig. 1A-C). Representative data of three independent experiments were shown in Fig. 2A.

As shown in Response Fig. 1A-C, cells infected with the $\Delta ospC1$ mutant showed significantly higher caspase-8 activity than those infected with WT *Shigella*. By contrast, cells infected with $\Delta ospI$ showed significantly lower caspase-8 activity than those infected with WT *Shigella* infection (Response Fig. 1A-C).

(A-C) HT-29 cells were infected with the indicated *Shigella* strains and incubated for 8 h. Caspase-8 activity was measured and is reported as relative light units (RLU) of infected samples normalized to the value in uninfected samples. Data are expressed as the mean \pm SD from triplicate wells. * $P < 0.05$ (two-way ANOVA). Results of repeated three independent experiments were shown in (A), (B), and (C), respectively. (D) Pooled data from three independent experiments performed in triplicate (A-C). n.s., not significant.

However, when data are pooled from three independent experiments performed in triplicate, there was no significant difference between WT and $\Delta ospC1$ or WT and $\Delta ospI$ (Response Fig. 1D).

Taken together, these data indicates that it is not appropriate statistical analysis to pool data from three independent experiments and average the results of all experiments because absolute values vary between experiments.

Therefore, we decided to show representative data of three independent experiments as described in many other report (Broz et al. 2012 Nature [PMID: 22895188]; Kayagaki N et al. 2015 Nature [PMID: 26375259]; Mitchell et al. 2020 Elife [PMID: 33074100]; Li et al. 2021 Nature [PMID: 34671164]; Ma et al. 2021 EMBO J [PMID: 34296442]; Peng et al. 2022 Mol Cell [35338844]; Ren et al. 2023 EMBO J [PMID: 37646198]).

We hope our explanation will be satisfactory.

2. In cases where there are 2 factors, e.g., Fig 1D, 1F, 2D, 3B, 3E, 3I etc (and similar ones elsewhere) - the authors have used Student's t tests, which is not correct. These data (averages from independent repeats, not technical repeats) should be analysed as 2-way ANOVAs. Student's t-tests are only appropriate when there are exactly 2 groups in the entire experiment.

> We appreciate this important comment. To address this comment, we re-performed statistical analysis of our data. Please see revised Figure Legends (line 955-1227). Statistical analysis was performed in GraphPad Prism. Differences between two groups were evaluated using unpaired two-tailed Student's *t*-test. One-way or two-way ANOVA was used to analyze differences among multiple groups. None of these changes in the statistical analyses affected our results or conclusion.

We hope you agree that the revised version of the manuscript is much improved.

3. It is not clear what stats were used for 1D, and some others, please add details.

> To address this comment, we added detailed information in revised Figure Legends (line 965-969).

All editorial and formatting issues were resolved by the authors.

Dear Dr. Ashida,

Congratulations on an excellent manuscript! I am very pleased to inform you that it has been accepted for publication in The EMBO Journal. Thank you for comprehensively addressing the initially raised referee concerns and all editorial requests for corrections and changes.

If you have any questions, please do not hesitate to contact the Editorial Office. Thank you for your contribution to The EMBO Journal. Working with you has been a pleasure.

Best regards,

Ioannis
